



# Top-down quantification of $NO_x$ emissions from traffic in an urban area using a high resolution regional atmospheric chemistry model

Friderike Kuik[1,2], Andreas Kerschbaumer[3], Axel Lauer[4], Aurelia Lupascu[1], Erika von Schneidemesser[1], and Tim M. Butler[1,2]

[1]Institute for Advanced Sustainability Studies, Potsdam, Germany
[2]Freie Universität, Fachbereich Geowissenschaften, Institut für Meteorologie, Berlin, Germany
[3]Senatsverwaltung für Umwelt, Verkehr und Klimaschutz Berlin, Germany
[4]Deutsches Zentrum für Luft- und Raumfahrt (DLR), Institut für Physik der Atmosphäre, Oberpfaffenhofen, Germany

*Correspondence to:* Friderike Kuik (friderike.kuik@gmail.com)

**Abstract.** With $NO_2$ limit values being frequently exceeded in European cities, complying with the European air quality regulations still poses a problem for many cities. Traffic is typically a major source of $NO_x$ emissions in urban areas. High resolution chemistry transport modelling can help to assess the impact of high urban $NO_x$ emissions on air quality in and outside of urban areas. However, many modelling studies report an underestimation of modelled $NO_x$ and $NO_2$ compared with observations. Part of this model bias has been attributed to an underestimation of $NO_x$ emissions, particularly in urban areas. This is consistent with recent measurement studies quantifying underestimations of urban $NO_x$ emissions by current emission inventories, identifying the largest discrepancies when the contribution of traffic $NO_x$ emissions is high. This study applies a high resolution chemistry transport model in combination with ambient measurements in order to assess the potential underestimation of traffic $NO_x$ emissions in a frequently used emission inventory. The emission inventory is based on officially reported values and the Berlin-Brandenburg area in Germany is used as a case study. The WRF-Chem model is used at a 3 km x 3 km horizontal resolution, simulating the whole year 2014. The emission data are downscaled to a resolution of 1 km x 1 km. An in-depth model evaluation including spectral decomposition of observed and modelled time series and error apportionment suggests that an underestimation in traffic emissions is likely one of the main causes of the bias in modelled $NO_2$ concentrations in the urban background, where $NO_2$ concentrations are underestimated by ca. 8 µg m$^{-3}$ (-30%) on average over the whole year. Furthermore, a diurnal cycle of the bias in modelled $NO_2$ suggests that a more realistic treatment of the diurnal cycle of traffic emissions might be needed. Model problems in simulating the correct mixing in the urban planetary boundary layer probably play an important role in contributing to the model bias, particularly in summer. Also taking into account this and other possible sources of model bias, a correction factor for traffic $NO_x$ emissions of ca. 3 is estimated for weekday daytime traffic emissions in the core urban area, which corresponds to an overall underestimation of traffic $NO_x$ emissions in the core urban area of ca. 50%. Sensitivity simulations for the months of January and July using the calculated correction factor show that the weekday model bias can be improved from -8.8 µg m$^{-3}$ (-26%) to -5.4 µg m$^{-3}$ (-16%) in January on average in the urban background, and -10.3 µg m$^{-3}$ (-46%) to -7.6 µg m$^{-3}$ (-34%) in July. In addition, the negative bias of weekday $NO_2$ concentrations downwind of the city in the rural and suburban background can be reduced from -3.4 µg m$^{-3}$ (-12%) to -1.2 µg m$^{-3}$ (-4%) in January and -3.0 µg m$^{-3}$ (-22%) to -1.9 µg m$^{-3}$ (-14%) in July. The results and their consistency with



findings from other studies suggest that more research is needed in order to more accurately understand the spatial and temporal variability in real-world $NO_x$ emissions from traffic, and apply this understanding to the inventories used in high resolution chemical transport models.

## 1 Introduction

Limit values for ambient $NO_2$ concentrations (Ambient Air Quality Directive 2008/50/EC) as well as $NO_x$ exhaust emission standards are set by European legislation, but ambient measurements show that $NO_2$ concentrations still frequently exceed the European annual mean limit value of 40 µg m$^{-3}$ (Eea, 2016; Minkos et al., 2017). For example, 12% of all measurement sites in Europe registered exceedances of the annual mean limit value in 2014, most of them located at the roadside. Within Europe, Germany had the highest median $NO_2$ concentrations in 2014 (Eea, 2016), where it was estimated that the limit value was exceeded at 57% of all traffic sites (Minkos et al., 2017). While there is a downward trend in $NO_2$ concentrations, extrapolating the current trend to 2020, exceedances are still expected at 7% of the stations in 2020, requiring additional measures in order for the European air quality goals to be met (Eea, 2016).

In general, traffic is the most important source of $NO_x$ emissions in Europe contributing 46% in 2014 in the EU-28, with considerably higher contributions to ambient $NO_2$ concentrations in urban areas (Eea, 2016). $NO_x$ emissions from diesel vehicles, the main traffic $NO_x$ source, have recently been a strong focus of international media attention: despite increasingly strict emission standards for diesel cars with the introduction of the Euro 5 and Euro 6 norms, in real-world driving conditions Euro 5-certified cars exceed the emission limit of 0.18 g km$^{-1}$ by an average factor of 4-5 (e.g. Eea, 2016; Hausberger and Matzer, 2017) and the newer Euro 6-cars exceed the emission limit of 0.08 g km$^{-1}$ by an average factor of 6-7 (e.g. Eea, 2016; Briefing, 2016).

$NO_x$ impacts human health, ecosystems and climate directly and indirectly as precursor of tropospheric ozone ($O_3$) and particulate matter (PM). Health impacts of $NO_2$ include adverse respiratory effects (Who, 2013), and the effect of road traffic $NO_2$ on premature mortality might be more than ten times larger than the effect of road traffic PM2.5 (Harrison and Beddows, 2017).

In order to support policy makers in identifying suitable measures to reduce roadside and urban background $NO_2$ concentrations to levels well below the limit value, as well as for assessing the health impact of current and future $NO_2$ concentrations, air pollution modelling is a valuable tool (e.g. Von Schneidemesser et al., 2017). Chemistry transport models can be used to assess the impact of local emissions on air chemistry and air quality in the surroundings and downwind of the emission sources. Online-coupled models, such as the chemistry version of the Weather Research and Forecasting model (WRF-Chem, Grell et al., 2005), have several advantages compared with offline approaches. These include, for example, a numerically more consistent treatment and a more realistic representation of the atmosphere, particularly in case of high model resolution (Grell and Baklanov, 2011).

Due to its short lifetime in the atmosphere, $NO_2$ is more spatially variable than for example $O_3$, particularly in urban areas with locally high $NO_x$ emissions. This is one of the reasons why models with higher spatial resolutions of a few km are





capable of representing observed NO$_2$ concentrations better than coarser models, with better performance if emission input and meteorological data are also available at the high resolution (e.g. Schaap et al., 2015). In terms of model evaluation, comparing NO$_2$ concentrations averaged over a coarse model grid cell with point measurements can lead to mismatches (Solazzo et al., 2017), with a better comparability achieved through high model resolutions of only a few km or less, depending on the size of the city. Simulating air quality in Mexico City, Tie et al. (2010) showed that reasonable model results can be achieved at a ratio of city size to model resolution of ca. 6:1.

However, many modelling studies report discrepancies between modelled with observed NO$_2$ concentrations, which are in parts attributed to an underestimation of traffic NO$_x$ emissions. All but one model simulating the European domain during the model intercomparison project AQMEII phase 2 underestimate annual mean NO$_2$ concentrations by 9%-45% on average. Some of them overestimate NO$_2$ concentrations at nighttime (Im et al., 2015), meaning that daytime concentrations are underestimated even more than the average model bias would indicate. Similarly, the European models contributing to the more recent AQMEII phase 3 intercomparison show an under-prediction of NO$_2$ concentrations throughout the whole year, with the sole exception of one model (Solazzo et al., 2017). In the Eurodelta model intercomparison study (Bessagnet et al., 2016), the participating models simulate NO$_2$ concentrations reasonably well on average compared with observations in the rural background, but most models show an underestimation of daytime NO$_2$ on average, particularly in summer (Fig. 9 from Bessagnet et al., 2016). Few studies focus particularly on NO$_2$ in urban areas: Terrenoire et al. (2015) simulated air quality over Europe at a horizontal resolution of 0.125° x 0.0625° with the CHIMERE model for 2009 and found that NO$_2$ concentrations are underestimated by more than 50% in urban areas. Schaap et al. (2015) show that the bias in modelled NO$_2$ concentrations in urban areas is reduced with increasing model resolutions, but still report negative biases for a model resolution of 7 km x 7 km, between 6 and 10 µg m$^{-3}$ for different offline-coupled chemistry transport models. Degraeuwe et al. (2016) assess the impact of different diesel NO$_x$ emission scenarios on air quality in Antwerp, basing street canyon modelling on urban background concentrations modelled with LOTOS-EUROS at a horizontal resolution of ca. 7 km x 7 km. They report a low bias in modelled urban background NO$_2$ concentrations of ca. 20%, requiring bias correction for the further analysis of the emission scenarios. Kuik et al. (2016) evaluated air quality simulated with WRF-Chem over the Berlin-Brandenburg region and found underestimations of NO$_2$ concentrations at daytime, and overestimations at nighttime.

Many studies attribute an underestimation of observed NO$_2$ concentrations to an underestimation of emissions (e.g. Solazzo and Galmarini, 2016; Degraeuwe et al., 2016; Giordano et al., 2015) and particularly traffic emissions in urban areas (Terrenoire et al., 2015). Further reported causes of the disagreement include problems with simulating the correct PBL height and mixing in the model (e.g. Solazzo et al., 2017; Kuik et al., 2016).

Modelling studies for North America report lower negative or even positive biases in modelled NO$_2$ concentrations (e.g Solazzo et al., 2017). While total NO$_x$ emissions reported for Europe are on average already larger than for North America by a factor of more than 2 (Im et al., 2015), these differences might indicate an even larger contribution of diesel car emissions to measured NO$_2$ concentrations, as the share of diesel cars is a main difference in emission sources between Europe and North America. Thus, large differences in the model bias between Europe and North America would be consistent with an underestimation of diesel traffic emissions in Europe.





Emissions are typically estimated from a combination of activity data and emission factors. Emission factors for road transport emissions depend on the fuel type and the car type (heavy duty or light duty, exhaust treatment) as well as on the driving conditions including road type and speed (e.g. Hausberger and Matzer, 2017). While activity data are only assumed to have an uncertainty of ca. 5%-10%, the emission factor is more difficult to quantify in many cases (Kuenen et al., 2014, and references therein). Emission factors for road transport, for example, may have an error range between 50% and 200% (Kuenen et al., 2014). Recent studies for London show that $NO_x$ emissions from flux measurements are up to 80% (Lee et al., 2015), or a factor of 1.5-2 (Vaughan et al., 2016) higher than $NO_x$ emissions from the UK National Atmospheric Emissions Inventory, with largest discrepancies found in cases where traffic is the dominant source of $NO_x$ concentrations. Karl et al. (2017) conclude from eddy covariance measurements in Austria that traffic related $NO_x$ emissions in emission inventories frequently used by air quality models can be underestimated by up to a factor of 4 for countries where diesel cars represent a major fraction of the vehicle fleet and have a significant contribution to reported biases in modelled $NO_2$ concentrations.

In this study the aim is to quantify the underestimation of traffic emissions in a widely used state-of-the art emission inventory based on officially reported emissions, for simulating $NO_2$ concentrations in an urban area with high resolution. We use the Berlin-Brandenburg area as a case study, and use the WRF-Chem model to simulate $NO_2$ concentrations. The model, model simulations and input data are described in Sect. 2, and observational data used are described in Sect. 3. The emission inventory used here is the TNO-MACC III inventory (Kuenen et al., 2014), downscaled to ca. 1 km x 1 km for the Berlin-Brandenburg area (Kuik et al., 2016), also described in Sect. 2. The analysis builds on advanced model evaluation techniques including an operational and a diagnostic evaluation (outlined in Sect. 4) of the modelled $NO_2$ concentrations (Dennis et al., 2010), with the aim of assessing the contribution of different sources of model error (Sect. 5). Based on this analysis, a correction factor for traffic emissions is calculated, and additional sources of the model bias are discussed (Sect. 6). The factor is then tested in two sensitivity simulations for January and July 2014. In addition, we analyse observational data of $NO_2$ concentrations and traffic counts, assessing the linear scaling assumed between emissions and traffic counts for the temporal distribution of emissions in chemistry transport models. Section 7 closes with a summary and conclusions from the results.

## 2 Model simulations

### 2.1 Model setup

We use the Weather Research and Forecasting model (WRF) version 3.8.1 (Skamarock et al., 2008), with chemistry and aerosols (WRF-Chem, Grell et al., 2005; Fast et al., 2006). The setup includes two model domains centred over Berlin, at horizontal resolutions of 15 km x 15 km and 3 km x 3 km, using one-way nesting. The model top is at 50 hPa, using 35 vertical levels with the first model layer at approximately 30m above the surface. There are 12 levels in the lowest 3km. Urban processes (meteorology) are parametrized with the single layer urban canopy model, with input parameters specified for Berlin as described in Kuik et al. (2016) and three urban land use categories. The setup further includes the RADM2 chemical mechanism with the Kinetic Pre-Processor (KPP) and the MADE/SORGAM aerosol scheme. The MOZART chemical mechanism is used in a sensitivity test. All physics and chemistry schemes used in this study are listed in Table 1.





Small changes in the code have been made. The initialization of the dry deposition (module_dep_simple.F) has been adapted in order to account for three urban land use categories as described in Kuik et al. (2016) and references therein. Nighttime mixing over urban areas is not accounted for sufficiently by the urban parametrization and the PBL scheme and thus adjusted (dry_dep_driver.F) as described in the Supplementary Material.

## 2.2 Model input data

We use the European Centre for Medium-Range Weather Forecasting (ECMWF) Interim reanalysis (ERA-Interim, Dee et al., 2011) with a horizontal resolution of 0.75°x0.75°, and a temporal resolution of 6h, interpolated to 37 pressure levels (with 29 levels below 50 hPa) as meteorological initial and lateral boundary conditions. The sea surface temperature is updated every 6 hours. The data are interpolated to the model grid using the standard WRF pre-processing system (WPS). Chemical boundary conditions for trace gases and particulate matter are created from simulations with the global chemistry transport Model for Ozone and Related chemical Tracers (MOZART-4/GEOS-5 Emmons et al., 2010). Instead of the standard USGS land use data we use CORINE data (EEA, 2014), remapped to the USGS classes, using three categories characterizing the urban area. This provides are more realistic characterization of the land use in the Berlin-Brandenburg area (Kuik et al., 2016; Churkina et al., 2017). The emission input data and its pre-processing are described in Sect. 2.4.

## 2.3 Simulation procedure

Following the work flow used in AQMEII phase 2 (Brunner et al., 2015), we re-initialize the simulation every two days, with a one day spin-up of the model meteorology. To ensure consistency in the chemical fields, we start each new two-day simulation from the chemistry fields of the previous simulation. For the base run using the RADM2 chemistry scheme we do a full year (2014) simulation. The results of this simulation are used to derive a correction factor for road traffic emissions, as explained in Section 6.1. For computational reasons, the simulation is divided into two parts covering the first six and the last six months of the year. Both simulations are preceded by a spin-up period of 4 days. We do a one-month sensitivity simulation with the MOZART chemistry scheme (July 2014), and two sensitivity simulations with increased traffic emissions for January and July 2014, all with the same simulation procedure. All model simulations are listed in Table 2.

## 2.4 Emissions

### 2.4.1 General description

The emission data used in this study are from the TNO-MACC III inventory (Kuenen et al., 2014). The latest available year is 2011, which we use for simulating the year 2014. Details on the emission inventory and the way these emissions are used in the present WRF-Chem setup can be found in Kuik et al. (2016) and references therein, and are briefly summarised here. The data are originally at a horizontal resolution of ca. 7 km x 7 km, which we downscale for the Berlin-Brandenburg region based on proxy data (Fig. 1). As it has been shown that downscaling the emission data to the resolution of the model grid helps to better capture the spatial distribution of air pollutant concentrations, we updated the downscaling procedure (see Supplementary



Material). The updates include an extension of the region for which the emissions are downscaled from Berlin to the whole Berlin-Brandenburg region. In addition, we only downscale those emission categories (SNAP categories) which are both of main interest for studying $NO_2$ in an urban area and also represented well by the proxy data chosen. This ensures that we are not suggesting a higher precision than achievable with the available proxy data.

### 2.4.2 Emission processing

Kuik et al. (2016) concluded that when simulating urban air quality with high resolution and using emission input data at high resolution, a more detailed treatment of the vertical distribution of point source emissions might further improve the model results. For this reason in this study, the emissions are distributed vertically based on profiles adapted from Bieser et al. (2011), i.e. emissions from the energy industry are distributed between the third and seventh model layer, emissions from other industrial sources as well as from the extraction and distribution of fossil fuels are distributed between the first four model layers, waste treatment emissions are distributed in the first five model layers and airport emissions (LTO cycle) are distributed vertically into the first seven layers (see Supplementary Material for further details).

TNO-MACC III emissions are provided as annual totals. For each emission (SNAP) category separately, we apply factors distributing the emissions for each month, day of the week (weekend vs. weekday), and hour of the day (diurnal cycle) based on Builtjes et al. (2002), with the exception of the diurnal cycle of traffic emissions. Previous studies highlighted the importance of using locally available information when specifying temporal profiles of emissions (e.g. Mues et al., 2014). Here we apply a diurnal cycle of traffic emissions (fraction of total daily emissions per hour of the day) calculated based on traffic counts provided by the Berlin Senate Department for the Environment, Transport and Climate (data used from 2007-2016) and by the German Federal Highway Research Institute BASt (Bundesanstalt für Straßenwesen, 2017, data used from 2003-2016). The diurnal cycle applied here is obtained by calculating the fraction of average daily traffic counts in Berlin at each hour of the day, thus assuming a linear scaling of traffic emissions with traffic counts as also assumed by Builtjes et al. (2002). Following Builtjes et al. (2002), we apply a uniform diurnal cycle for each day of the week, making no distinction between weekends and weekdays. The main differences between the profiles calculated based on locally available information and the hourly emission factors from Builtjes et al. (2002) include an earlier increase of traffic emissions in the morning by ca. one hour and more evenly distributed high traffic emissions during the day with less pronounced morning and afternoon peaks.

$NO_x$ is emitted mainly as NO, but also includes a fraction directly emitted as $NO_2$ (the primary $NO_2$ fraction, f-$NO_2$) by combustion engines. Here, $NO_x$ is emitted as NO for all SNAP categories except "road transport" and "non-road transport". For non-road transport and all road transport emissions except diesel, $NO_x$ is emitted as 10% $NO_2$ and 90% NO. Road transport diesel emissions include both light duty vehicle (LDV) and heavy duty vehicle (HDV) emissions. For the latter, we also assume a f-$NO_2$ of 10%, while for light duty vehicles we assume a f-$NO_2$ of 26% (Carslaw, 2005). Combining this with the TNO-MACC share of diesel emissions attributable to LDV (43%) and HDV (57%), we obtain a combined f-$NO_2$ for road transport diesel $NO_x$ emissions (SNAP 72) of 17%. Test simulations varying the f-$NO_2$ for diesel LDV between 10% and 55% have shown that the simulated NO, $NO_2$ and $NO_x$ concentrations have very little sensitivity towards the f-$NO_2$ of LDV diesel emissions, while small differences in the simulated ozone concentrations were seen. As further sensitivity simulations on this





topic are beyond the scope of this study and differences were small, we chose to use a f-NO$_2$ that was around the mid-point of those values documented for LDV diesel NO$_x$ emissions (26%).

### 2.4.3 Comparison of the downscaled TNO-MACC III emissions with a local inventory

Local NO$_x$ emissions from road transport are available for 2009 (Berlin Senate Department for the Environment, Climate and
Transport, online). In comparison with the downscaled TNO-MACC III emissions for the Berlin grid cells (2011), traffic NO$_x$ emissions from the local inventory are 6% higher. The geographical distribution of the emissions in the local inventory is very similar to the downscaled version of TNO-MACC III used in this study (Fig. 1).

## 3   Observational data

### 3.1   AirBase observations and NO$_2$ uncertainty

NO$_2$, NO$_x$ and O$_3$ measurements are taken from AirBase (Eea, 2017), a database compiling air quality observations from the EU Member States and associated countries, performed as required by EU clean air legislation. In the case of Germany the measurements are performed by the federal states. For the comparison with model results, observations from stations within Berlin and in the adjacent surroundings in the Federal State of Brandenburg representing "urban background", "suburban background" and "rural near-city" conditions are used (Fig. 2). For our analysis, we re-classify the AirBase station "DEBE066"
in Berlin-Karlshorst from "urban background" to "suburban background", as the station is not located in the core area of the city and pollutant concentrations measured there are similar to concentrations measured at other suburban background stations. As a result, four stations for each classification type are used in this study: Amrumer Straße (DEBE010), Brückenstraße (DEBE068), Belziger Straße (DEBE018) and Nansenstraße (DEBE034) in the urban background, Blankenfelde-Mahlow (DEBB086), Buch (DEBE051), Groß Glienicke (DEBB075) and Johanna und Willy Brauer Platz (DEBE066) in the suburban background, and
Frohnau (DEBE062), Grunewald (DEBE032), Müggelseedamm (DEBE056) and Schichauweg (DEBE027) in the rural near-city background.

In addition, five measurement stations representing "traffic" conditions within Berlin, which are located next to major roads within the core area of the city, and assumed to be primarily influenced by traffic emissions, are used for the observation-based analysis (Sect. 6.3).

NO$_2$ concentrations used for this study were measured using chemiluminescence. With this method, NO$_2$ is converted to NO with a molybdenum converter before being detected using chemiluminescence. A limitation of this method is that other nitrogen-containing species (PAN, HNO$_3$) are also converted to NO in this process. In a comparison study, Steinbacher et al. (2007) found that only 73%-82% of the measured NO$_2$ with this method is "real" NO$_2$, at a rural background site in Switzerland. However, they state that reasonable results are obtained with this type of converter at urban background sites. Villena
et al. (2012) compared NO$_2$ concentrations in urban smog conditions in Santiago de Chile using chemiluminescence detection with a molybdenum converter and differential optical absorption spectroscopy and found large differences between measured



concentrations during daytime. Further sources of uncertainty are introduced in the detection itself, for which NO reacts with $O_3$, producing the luminescence signal to be detected. Gerboles et al. (2003) assess the uncertainty of $NO_2$ measurements, and Pernigotti et al. (2013) derive a simplified procedure in order to calculate the $NO_2$ measurement uncertainty, which we apply in order to obtain a rough estimate of the uncertainty range of $NO_2$. Accordingly, the uncertainty (u) of the observed $NO_2$

concentrations x at time i is quantified as follows:

$$u(x_i) = u_{rRV} \cdot \sqrt{(1-\alpha)x_i^2 + \alpha \cdot RV^2} \tag{1}$$

Here, urRV is an estimate of the relative uncertainty around a reference value RV, and $\alpha$ is the fraction of uncertainty not proportional to the reference value. We use the coefficients corresponding to the mean uncertainties of the individual parameters, i.e. urRV=0.09, $\alpha$=0.06 and the reference value RV=200 µg m$^{-3}$ (Pernigotti et al., 2013).

**3.2  Meteorological data**

In order to complement the analysis and to investigate potential influences of the modelled meteorology on modelled $NO_2$ concentrations, we include a comparison of modelled meteorology with observations. This includes observations of 2m temperature, and 10m wind speed and direction, all provided by the German Weather Service and available online (Kaspar et al., 2013). In addition, mixing layer height derived from ceilometer measurements at Nansentraße during the BAERLIN2014 cam-

paign (Geiß et al., 2017) are used for a qualitative comparison with the modelled mixing layer height (see Kuik et al., 2016, for a discussion of this type of comparison). The data are generally available between 20 June and 27 August 2014, but include a number of gaps.

## 4  Analysis and evaluation metrics

### 4.1  Analysis of model results

Modelled $NO_2$ concentrations are evaluated with the aim of using the model setup for policy-relevant analyses of urban $NO_2$ concentrations with high temporal and spatial resolution, and in order to identify the main sources of the errors in modelled $NO_2$ concentrations. For this, we use both operational and diagnostic evaluation metrics, which are explained in the following.

Operational evaluation metrics applied here are based on Thunis et al. (2012) and Pernigotti et al. (2013). They include an analysis of the mean bias (MB) and normalized mean bias (NMB), the correlation coefficient (R), and the root mean square

error (RMSE, as defined in the Supplementary Material). The model error is compared with the model quality objective (MQO) and performance criteria calculated from $NO_2$ observations and their uncertainty. Following Thunis et al. (2012) and Pernigotti et al. (2013), a MQO lower than 0.5 indicates that the model results are on average within the range of the measurement uncertainty, and further efforts to improve model performance are not meaningful. A MQO between 0.5 and 1 indicates that the uncertainties of model and observations overlap, and that the model might still be a better predictor of the true value

than the observations. A MQO greater than 1, on the other hand, indicates significant differences between the model and the




observations. The MQO is defined as follows:

$$\text{MQO} = \frac{1}{2} \frac{\text{RMSE}}{\text{RMS}_\text{U}} \tag{2}$$

With $\text{RMS}_\text{U}$ being the root mean square of the measurement uncertainty. The performance criteria for mean bias, normalized mean bias and correlation coefficient as defined in Pernigotti et al. (2013) are listed in the Supplementary Material. As the uncertainty of $NO_2$ measurements is partly concentration-dependent, the MQO and the other performance criteria differ between station classes and seasons.

The operational evaluation and model quality objectives are intended to support an assessment of the extent to which a model can be used for policy-relevant analyses, but do not point to the underlying processes that might lead to a disagreement between model results and observations. Furthermore, the calculation of the $NO_2$ measurement uncertainty underlying the calculation of the MQO and performance criteria is also based on a number of uncertain parameters.

We thus complement the analysis with a diagnostic evaluation, comparing the individual spectral components of the modelled and observed time series. This is done following Solazzo and Galmarini (2016) and Solazzo et al. (2017): we use a Kolmogorov-Zurbenko filter (Zurbenko, 1986), a widely used filter in the analysis of air quality data based on calculating the iterative moving average of a time series, in order to decompose the modelled and observed time series into contributions from different time scales. The Kolmogorov-Zurbenko filter is a low pass filter, with the length of the moving average window and the number of iterations determining the spectral component to be filtered. Taking the difference between two filtered time series (band-pass filter) makes it possible to decompose the observed and measured time series into an intra-diurnal component (ID, < 0.5 days), a diurnal component (DU, 0.5-2.5 days), a synoptic component (SY, 2.5-21 days) and a long-term component (LT, >21 days) with the property

$$\text{TS}(x) = \text{LT}(x) + \text{SY}(x) + \text{DU}(x) + \text{ID}(x). \tag{3}$$

Here, TS describes the full time series of the species x. This is described in detail in Solazzo et al. (2017) and Solazzo and Galmarini (2016) and references therein. Further detail is also given in the Supplementary Material.

By assessing the error of each component individually it is then easier to relate the error to the model process(es) characteristic at the respective time scale. The error analysis of the different spectral components is done by "error apportionment" (Solazzo et al., 2017), breaking down the mean square error (MSE) into bias, variance ($\sigma$) error and minimum achievable mean square error (mMSE) as follows:

$$\text{MSE} = (\text{mod} - \text{obs})^2 + (\sigma_\text{mod} - \text{r}\sigma_\text{obs})^2 + \text{mMSE} \tag{4}$$

As described by Solazzo and Galmarini (2016), the minimum achievable mean square error is determined by the observed variability that is not reproduced by the model. While this approach helps investigating the sources of model errors, it does not allow for clearly identifying or quantifying them as several processes take place on similar time scales, and because this filtering method does not allow for a complete separation of the different spectral components (see Solazzo et al., 2017, for a discussion of this issue).



In addition to this operational and diagnostic analysis of simulated $NO_2$ concentrations, we include a brief evaluation of selected key meteorological parameters (temperature, wind speed and direction) as well as further chemical species ($O_3$, $NO_x$) as WRF-Chem is an online-coupled model and $NO_2$ is tightly linked to NO and $O_3$.

### 4.2 Observation-based analysis

As traffic emissions are the focus of this study, the analysis of the model results is complemented with an analysis based on observations of roadside and urban background $NO_2$ concentrations and traffic counts. Like in many chemistry transport modelling studies, we assume a linear scaling of traffic emissions with traffic counts, which are used as a proxy for calculating time profiles of traffic emissions for each month, day of the week and hour of the day. While it has been shown that model results can be improved by taking into account country-specific driving patterns as well as by applying separate diurnal cycles for heavy and light duty vehicles (Mues et al., 2014), local traffic conditions (e.g. congestion) are currently not taken into account in the calculation of the diurnal cycles.

Using observations of traffic counts and roadside $NO_x$ concentrations in Berlin obtained at the same locations and times (data described in Sect. 2.4.2 and 3.1), we assess how much of the observed variance in $NO_x$ concentrations can be explained with traffic counts in a linear model. In addition to a linear fit, other types of relationships (e.g. quadratic, exponential) are also explored. We neglect other influences on observed $NO_x$ concentrations such as other emission sources and large-scale and local meteorological conditions. In order to account for different conditions at different hours of the day, we fit the data separately for each hour of the day. The intention of this analysis is not to build a statistical model for roadside $NO_x$ concentrations, but rather to give insight into the type of relationship between roadside $NO_x$ concentrations and traffic counts, complementing the model simulations done in this study.

## 5 Model evaluation

### 5.1 Meteorology

An in-depth evaluation of modelled meteorology obtained with a similar model setup is presented in Kuik et al. (2016) for the summer (JJA) of 2014. Here, model results for the whole year of 2014 are presented and discussed. Changes in the model setup compared with the setup presented in Kuik et al. (2016) are the planetary boundary layer scheme (MYNN, Nakanishi and Niino (2006) instead of YSU, Hong et al. (2006)) and re-initialization of the model meteorology every 2 days as described in Sect. 2. Tests showed that though the change in planetary boundary layer scheme did not introduce considerable improvements, it did seem to lead to a slightly better match of model results with observations in the timing of the decrease of the boundary layer in the evening. Here an additional brief model evaluation is done in order to ensure that the modelled meteorology still reproduces observations reasonably well.

Modelled and observed temperature and wind speed are compared at five stations run by the German Weather Service, including Schönefeld, Tegel and Tempelhof in Berlin and Lindenberg and Potsdam outside of Berlin (Table 3). Across the





stations, annual mean temperature is simulated well, with mean biases smaller than -1°C outside of Berlin and just above -1°C within Berlin. Modelled and observed hourly temperatures correlate well with R=0.96 at all five stations. Small seasonal differences exist, with somewhat higher biases in winter (as large as -1.7°C in Tegel) and somewhat lower biases in spring (e.g. -0.1°C in Schönefeld). Annual mean wind speed is somewhat overestimated within Berlin (between 0.02 m/s and 0.45 m/s,

or up to 13%), with correlations of the hourly values between 0.74 and 0.78 within Berlin. In winter, wind speed is slightly underestimated at two out of the three stations within Berlin (-2% and -7% at Tegel and Schönefeld, respectively), while it is overestimated somewhat more in spring and summer (up to 0.58 m/s, or ca. 20% in Tegel). In spring and summer, the main wind directions are captured relatively well by the model (see Fig. S2 and S3 in the Supplementary Material). In autumn, wind from the east, the main wind direction, is modelled less frequently than observed, but wind from the south-east is modelled too

frequently compared with observations. In winter, modelled wind comes from south and south-west too frequently compared with observations, at the expense of south-easterly wind directions, as depicted in Fig. S2 and S3. Compared with Kuik et al. (2016), an improvement in summer mean bias in wind speed is seen; with the JJA mean bias between 0.3 and 0.4 m/s smaller than that of the comparable simulation in Kuik et al. (2016) at all Berlin stations, and JJA correlation coefficients improved by ca. 0.1. This can probably be attributed to the continuous re-initialization of modelled meteorology in this simulation.

In addition, modelled and ceilometer-derived mixing layer heights (MLH) are compared (Fig. S4 in the Supplementary Material). Even though a quantitative comparison between the modelled MLH and the MLH height derived from optical measurements is difficult to interpret (see Kuik et al., 2016), a qualitative comparison of mean diurnal cycles gives insight into the timing of the deepening of the MLH. The comparison shows that the modelled increase of the summer MLH in the morning is too early, already starting at ca. 4 am in the model. Though the precise time of the observed MLH increase cannot

be determined from the available data, it takes place between 5am and 7am (Fig. S4 in the Supplementary Material). An early modelled deepening of the mixing layer might lead to a too early and thus too strong mixing of chemical species in the model.

**5.2   Operational evaluation of simulated chemical species**

Seasonally and station-class averaged performance metrics are listed in Table 4 for $NO_2$, $NO_x$ and $O_3$. $NO_2$ and total $NO_x$ are biased low throughout the seasons and station classes, with the highest (absolute and relative) mean biases for urban background

stations both annually and seasonally. The model bias is relatively low at rural and suburban background stations, with annual mean biases of only up to -2.8 µg m$^{-3}$ (-19%). Correlation coefficients of modelled with observed hourly concentrations are R=0.50 and R=0.55 in the rural and suburban background, respectively.

$NO_2$ at urban background sites is biased by -7.8 µg m$^{-3}$ (-29%) on average, with a higher negative bias in spring (-10.2 µg m$^{-3}$, -38%) and summer (-9.3 µg m$^{-3}$, -41%) and smaller negative biases in autumn (-4.9 µg m$^{-3}$, -17%) and winter (-6.8

µg m$^{-3}$, -22%). Modelled hourly concentrations correlate reasonably well with observations in autumn, spring and winter (R between 0.51 and 0.55), but worse in summer (0.36).

Modelled hourly ozone concentrations correlate reasonably well with observations at all station classes throughout the whole year (R between 0.70 and 0.73), but with lower correlations for individual seasons. This shows that intra-seasonal differences are represented well by WRF-Chem, with slightly worse representations of inter-seasonal variations. Modelled



ozone concentrations are biased high at most stations and in most seasons, with the exception of a low bias in summer in the urban background.

For NO$_2$, the MQO (Eq. 2) is greater than 0.5, but smaller than 1, both annually averaged and in all seasons at rural near-city background and suburban background stations. For urban background sites the MQO is larger than 1 both on annual average

and in spring and summer, and just below 1 in autumn and winter, emphasizing that the model performs reasonably well in the rural and suburban background, but the disagreement between model results and observations is larger in the urban background. This suggests that processes or emissions typical for urban areas are an important source of model error.

In order to test the sensitivity of the results to the selected chemical mechanism, we compare modelled NO$_2$ and total NO$_x$ concentrations for July with two different chemical mechanisms: RADM2 (the base configuration in this study) and MOZART.

For all station classes in and around Berlin, the modelled NO$_x$ and NO$_2$ concentrations only show very small mean differences of -0.04 to -0.4 μg m$^3$ (NO$_x$) and -0.4 to -0.5 μg m$^3$ (NO$_2$, RADM2 - MOZART). This suggests that the model bias in NO$_2$ and total NO$_x$ concentrations of the base configuration is not strongly influenced by the choice of chemical mechanism, but rather results from other sources of error.

### 5.3  Diagnostic evaluation of simulated NO$_2$ concentrations

In order to further assess the model performance and identify main sources of the model bias, a diagnostic evaluation is done, by spectrally decomposing the modelled and observed time series of NO$_2$ and analysing the type of error of each component.

Averaging the decomposed time series over each station class, the modelled long term (LT) and synoptic (SY) components as defined in Sect. 4.1 correlate well with the observations: the correlation coefficient for the LT component is 0.83, 0.81 and 0.72 for rural near-city, suburban and urban background, respectively, and 0.60, 0.63 and 0.65 for the SY component (Fig. 3).

This suggests that changes on time scales of ca. 2.5 days to a few weeks are captured relatively well by WRF-Chem, which includes for example the modelled synoptic (meteorological) situation and is consistent with the good model performance in simulating observed meteorology. The correlation coefficients for the diurnal (DU) component are smaller, with 0.45, 0.52 and 0.48 for rural near-city, suburban and urban background, respectively. This suggests that the model has more difficulties in capturing variations at time scales of a few hours to 2.5 days than on longer time scales. This might be related to the diurnal

variations in modelled mixing, but also to the diurnal cycle of emissions. Particularly the latter is strongly influenced by traffic emissions in the urban area and might also point to deviations of the model-prescribed diurnal cycle in emissions from the real-world diurnal cycle.

With the procedure used for spectrally decomposing the NO$_2$ time series, the LT component is the only systematically biased component, with the other components fluctuating around zero. Decomposing the model error shows that the bias of

the LT component has the largest contribution to the error for urban background stations (ca. 30%, Fig. 4). NO$_2$ has a short life time and is mainly influenced by local and regional sources. This means that the boundary conditions are not likely to be a strong source of error. The negative bias in the LT component is consistent with both problems in daytime vertical mixing and an underestimation of emissions. As discussed in Sect. 3.1, NO$_2$ concentrations detected with chemiluminescence using





a molybdenum converter might be biased high due to interferences with other nitrogen-containing species (e.g. PAN, $HNO_3$) and could further contribute to discrepancies between modelled and observed $NO_2$ concentrations.

The second largest error at urban background stations and the largest error at rural near-city and suburban background stations is the mMSE of the diurnal component. This means that part of the observed variability is not reproduced by the model and is consistent with the comparably lower correlation coefficients of the diurnal component compared with the synoptic and long term components. Solazzo et al. (2017) relate this error to problems in comparing single point measurements with model grid cell values (incommensurability) and a disagreement in timing of modelled and observed concentrations, amongst others. The incommensurability can, in the case of $NO_2$, come from $NO_2$ observations being influenced by local sources that cannot be captured by WRF-Chem. The temporal variation of modelled $NO_2$ concentrations, in case of the diurnal component, can be influenced by the temporal profiles prescribed to the emission input data. Thus, the error is consistent with problems in the prescribed diurnal cycles of emissions including traffic emissions, but might also be related to a diurnally varying bias in emissions.

At rural near-city background stations, there is a relatively large contribution of the variance error of the diurnal component. This is probably caused by an overestimation of the standard deviation of observed diurnal components in autumn (Fig. S5 in the Supplementary Material), particularly pronounced at the site Frohnau in the north or Berlin, slightly west of the main emission sources. This might be explained by the disagreement in modelled and observed wind direction in autumn, leading to higher than observed $NO_2$ peaks in the model.

Solazzo et al. (2017) present a diagnostic model evaluation of the AQMEII phase 3 model simulations for the year 2010 and report the largest error of modelled $NO_2$ in winter, both for the European and North American domains simulated in AQMEII. Our results show the opposite for urban background stations (Fig. S5 in the Supplementary Material): the model error, and particularly the bias, is smallest in autumn and winter. While Solazzo et al. (2017) attribute the winter bias to a potential underestimation in residential combustion emissions, these seem to be captured comparably well by the TNO-MACC III inventory in the case of Berlin. The re-distribution of these emissions based on population density may also have contributed to a better spatial representation in our study.

## 5.4 Diurnal and weekly variation of the model bias

The results from the operational and diagnostic evaluation of modelled $NO_2$ concentrations suggest that emissions within the urban area are a main source of model error, both contributing to the model bias and the lower correlation with observations. Traffic emissions have the largest contribution to urban $NO_x$ emissions. As traffic emissions have a distinct weekly and diurnal cycle, we additionally assess mean diurnal cycles of modelled and observed $NO_2$ concentrations as well as the differences between weekdays and weekends. This also helps to further assess the contribution of problems in modelled mixing to the model error. In addition, we analyse the MQO and performance criteria separately for weekends (Saturday and Sunday) and weekdays (Monday through Friday). Public holidays that fall on a weekday are excluded from this analysis, as they were not treated separately from regular weekdays in the emission processing.



The comparison of mean modelled and observed $NO_2$ diurnal cycles shows distinct differences between station classes and weekend and weekday diurnal cycles (Fig. 5). The diurnal cycle of observed $NO_2$ concentrations is modelled reasonably well for rural and suburban background stations. In particular, nighttime concentrations are simulated well for rural and suburban background stations, and mostly underestimated in the urban background. Other WRF-Chem modelling studies often report

too little mixing at nighttime over urban areas leading to a strong overestimation of observed concentrations. In this study as in other modelling studies using WRF-Chem (Ravan Ahmadov, pers. comm.), a modification of the model code was applied in order to increase nighttime mixing. This, in combination with a more realistic vertical distribution of point source emissions (as described in Section 2.4.2), seems to improve model performance for $NO_2$ during nighttime. In addition, tests revealed that this change to the model code does not impact modelled daytime concentrations.

During weekdays, there is an underestimation of the observed morning peak in all seasons and at all station classes. Weekend diurnal cycles are modelled well at rural and suburban background stations. At urban background stations there is a larger disagreement between modelled and observed concentrations throughout the whole day on both weekends and weekdays. The underestimation of daytime urban background $NO_2$ concentrations is particularly strong in summer and spring. This might be explained by mixing over urban areas during daytime that is too efficient, caused for example by a turbulent diffusion

coefficient that is too large during daytime over urban areas in the lowest model layer. Other modelling studies have reported similar problems, reducing the coefficient over urban areas (e.g. of the CHIMERE model setup used in Schaap et al., 2015). An onset of the deepening of the boundary layer that is too early (Sect. 5.1) might further contribute to the disagreement in the modelled and observed morning peaks. Overall, this discussion shows that the representation of vertical mixing over urban areas might have to be improved to be physically more consistent in regional models, for example by better taking into

account urban heat and momentum fluxes and treating the urban parameterization consistently with chemistry. Measurements of vertical profiles of $NO_x$ in cities, particularly in the planetary boundary layer, would be helpful in order to evaluate the models.

The model underestimation of observed daytime $NO_2$ concentrations at urban background stations is stronger on weekdays than on weekends, and is particularly noticeable during the morning hours. This is consistent with an overall underestimation

of emission sources active in the morning hours on weekdays and potentially also a misrepresentation of the diurnal cycles of emissions in the model. Traffic emissions are distributed in the model throughout the day using a linear scaling with traffic counts (Sect. 2.4.2), which might fall short of accounting for relatively higher emissions during situations with high traffic and associated congestion. This issue is further assessed in Sect. 6.3.

Generally, throughout all seasons, the $NO_2$ MQO is not met on weekdays for urban background stations, but is smaller than

1 on weekends (Fig. 6). The pattern of the model-observation disagreement, and particularly the weekend-weekday differences, are consistent with traffic emissions as a main source of the bias, having a particularly large contribution to observed urban background concentrations.





## 6   Top-down quantification of NO$_x$ emissions from traffic

### 6.1   Calculation of a correction factor

The results from the operational and diagnostic evaluation of modelled NO$_2$ concentrations suggest that traffic emissions are
a main source of model error in the urban background: the bias and the mMSE of the diurnal component have the largest
contribution to the model error in the urban background throughout all seasons, which is consistent with both an underesti-
mation of the magnitude of traffic emissions, and a problem with their temporal distribution. This is further supported by the
smaller (absolute and relative) daytime bias of modelled NO$_2$ concentrations on weekends, where there is less traffic. In the
following, we derive a correction factor based on this model bias, which represents the degree to which traffic emissions are
underestimated in Berlin, but also takes into account that other sources of model error are likely to also contribute to this bias.

Besides biases in traffic emissions, problems in modelled mixing, which is particularly relevant in summer and spring when
the mixed layer is deeper than in other seasons, might contribute to the model bias. Other contributions to the NO$_2$ bias might
come from deviations of modelled from observed wind speed in certain periods, and a potential overestimation of NO$_2$ in
the observations by detection of other nitrogen containing compounds as discussed above. These sources of error are likely to
impact the model results equally on both weekends and weekdays, whereas an underestimation of traffic emissions will have the
largest impact on the results on weekdays. For the quantification of the underestimation of traffic emissions we assume that the
weekend bias is entirely caused by non-traffic-emission-related sources of error and thus use the difference between weekday
and weekend bias as an estimate for the traffic related bias. We use the weekday-weekend difference of the relative biases (Fig.
7), thus assuming that the model error due to other sources than traffic emissions roughly scales with the magnitude of modelled
concentrations. These are both conservative assumptions, as the correction factor would be much larger if the whole weekday
bias was regarded as caused by traffic emissions, and it would also be larger if the absolute weekday/weekend difference was
used.

In order to estimate the correction factor for traffic NO$_x$ emissions, we combine the weekday increment of the model bias as
defined above with the average fraction of NO$_x$ emissions from traffic to total NO$_x$ emissions in Berlin. The nighttime model
bias on weekends and weekdays at urban background stations is of similar magnitude on weekends and weekdays (Fig. 5). A t-
test shows that the differences between weekday and weekend bias are not statistically significant at a 95% confidence interval
after ca. 17:00 UTC and before ca. 5:00 UTC (depending on the season). Furthermore, traffic emissions used in the model
contribute only little to the total NO$_x$ emissions before 6:00 UTC. This suggests that an underestimation of traffic emissions
is only likely to have a significant contribution to the bias in modelled NO$_2$ concentrations between ca. 6:00 and 17:00 UTC.
Within the core area of the city where traffic is high (all areas within the "S-Bahn ring"/main core of the city), the average
contribution of traffic NO$_x$ to total NO$_x$ between 6:00 and 17:00 UTC is between ca. 30% and 55%, depending on the month
and hour of the day. Seasonal average values over the indicated time period are used for the calculation of the correction factor,
with 37% in winter, 47% in summer and 42% in autumn and spring.





With the above assumptions, we quantify the underestimation of traffic $NO_x$ emissions in the core urban area on weekdays between 6:00 and 17:00 UTC as follows, calculating a correction factor $F_{NOx}$:

$$f_{NOx} = \frac{1}{1+NMB} \cdot \frac{1}{s_t}, \tag{5}$$

With the (negative) $NMB = \frac{mod-obs}{obs}$, and $s_t$ denoting the traffic share of $NO_x$ emissions. Averaged over all urban background stations, and all seasons, as well as the time period between 6:00 and 17:00 this results in a correction factor of ca. 3. When averaged over all hours of the day, this factor corresponds to an overall underestimation of $NO_x$ traffic emissions in the urban centre by a factor of ca. 2, and an underestimation of all-source $NO_x$ emissions in the urban centre by a factor of ca. 1.5.

In order to gain more insight into the underestimation of the $NO_x$ emissions, we calculate a separate correction factor for each hour and season based on hourly mean seasonal biases and traffic $NO_x$ emission shares (Fig. S6 in the Supplementary Material). The seasonal correction factors show a small increase between 6:00 and 8:00 with a subsequent decrease, and then remain relatively constant from 11:00 to 17:00. The diurnal variations of the factors for the different seasons are qualitatively similar, and the factors vary in magnitude within a range of ca. 1 between the seasons, with the factors being larger in winter than in summer. The diurnal cycle of the correction factor could be due to a diurnally varying importance of other sources of the modelled $NO_2$ bias than the traffic emissions, such as mixing, but might also be due to a disagreement in the prescribed diurnal cycle of traffic emissions with the real-world diurnal cycle of traffic emissions. The seasonal differences can at least partly be explained with the seasonally varying relevance of other sources of model error, such as mixing, which has a bigger impact in summer and thus also leads to a bigger bias on the weekends, reducing the weekday increment. The seasonal differences might also be influenced by the temperature dependence of $NO_x$ emissions in newer diesel cars (Hausberger and Matzer, 2017), leading to higher $NO_x$ emissions at colder temperatures, which are not captured by the model.

Overall, the assumptions in these calculations are rather conservative: assuming the weekend bias is not caused by an underestimation of traffic emissions at all is likely to underestimate the effect of any traffic bias. As mentioned above, using the absolute weekday increment of the bias would also lead to higher correction factors. A further discussion of the model bias and correction factor looking into potential reasons contributing to an underestimation of traffic $NO_x$ emissions is presented in Sect. 6.4.

## 6.2 Sensitivity simulation with increased emissions

The weekday correction factor was applied to $NO_x$ traffic emissions for the core urban area of Berlin (within the "S-Bahn Ring") and tested in two sensitivity simulations for January and July 2014. The results (Table 5 and Fig. 8) show that the bias of modelled $NO_2$ concentrations at urban background stations decreases on average by 2.6 µg m$^{-3}$ (NMB decreases from -24% to -16%) in January, and by 2.0 µg m$^{-3}$ (from -43% to -34%) in July when applying the correction factors for $NO_x$ emissions from traffic. The decrease is larger when only considering weekdays, with a mean bias lower by 3.4 µg m$^{-3}$ (from -26% to -16%) in January and by 2.7 µg m$^{-3}$ (from -46% to -34%) in July. $NO_2$ concentrations on weekends are still represented reasonably well by the model in January (Fig. 8).The weekend bias is only changed (decreased) by lower than 0.4 µg m$^{-3}$ in both cases. Only a minor change would be expected, since emissions on the weekend are not changed in the sensitivity





simulations, compared to the base simulation. In January, the correlation of modelled with observed $NO_2$ concentrations in the urban background is improved by between 0.03 and 0.06 for urban background stations in the sensitivity simulation, but this is not the case in July (Table 5). The lack of improvement in the July correlation coefficient could be related to nighttime concentrations in July that seem to be very sensitive to the increase in emissions during daytime (Fig. 8, lower panel). Despite an

improved representation of nighttime concentrations compared to a previous study (Kuik et al., 2016), this sensitivity suggests the need for further attention to mixing processes in urban areas in high resolution chemistry transport models.

Bigger improvements are seen when comparing total $NO_x$: the mean bias for urban background stations is reduced from -16.4 to -10.3 µg m$^{-3}$ (NMB decreased from -35% to -22%) in January and from -11.1 to -8.1 µg m$^{-3}$ (from -45% to -33%) in July. Only considering weekday concentrations, these are improved by 8.1 and 3.9 µg m$^{-3}$ (from -37% to -12% and -48% to

-33%) in January and July, respectively. The differences in $NO_2$ and $NO_x$ improvements suggest that the impact of the primary $NO_2$ fraction in emitted $NO_x$ on observed and modelled concentrations might need to be assessed in greater detail.

While on average the normalised mean bias in modelled rural and suburban background is only reduced by 1-2% in both January and July, the simulation of $NO_2$ and $NO_x$ concentrations downwind of the city centre is improved considerably in the sensitivity simulation. For analysing the change in modelled downwind concentrations, the results are broadly divided based on

four main wind directions (N, W, S, E). For each wind direction bin, the results of two stations outside the core urban area are analysed, with Frohnau and Buch in the North, Johanna und Willi Brauer Platz and Mueggelseedamm in the East, Schichauweg and Blankenfelde-Mahlow in the South and Gross Glienicke and Grunewald in the West. Only situations with wind speeds above 2 m/s are considered. The statistics are calculated for stations where at least 72 hourly model-observation pairs exist in the respective wind speed bin, leaving 4 stations in January and 6 stations in July for the analysis, with between 91 and 228

model-observation pairs. With some differences between the stations, the bias of weekday downwind $NO_2$ concentrations was reduced by between ca. 1.5 and 3 µg m$^{-3}$ January and ca. 0.7 and 1.5 µg m$^{-3}$ in July. Thus, downwind $NO_2$ concentrations in the sensitivity simulation are only biased by ca. -4% (January) and -14% (July) on average (as compared to -12% and -22% in the base run). This shows that the increase in traffic emissions also helps improve modelled downwind concentrations.

Overall, in both January and July, the bias in modelled urban background $NO_2$ and $NO_x$ is improved but still negative.

Modelled downwind $NO_2$ concentrations are improved considerably, but with low negative biases remaining also in this case. The improvements are consistent with an underestimation of traffic emissions being a main source of error. However, the results also suggest that on the one side traffic emissions might still be too low, which is consistent with the correction factor being a rather conservative estimate. On the other side, a still negative bias is also consistent with other sources of error contributing considerably to the model-observation differences as discussed previously. A relatively large bias in July remains, consistent

with the mixing being an additional main source of error particularly in summer.

Modelled $O_3$ concentrations are not very sensitive to the changes in $NO_x$ concentrations. On average, modelled $O_3$ is reduced at urban background stations in January by 1.5 µg m$^{-3}$ (NMB decreases from 29% to 22%). In July, the increased $NO_x$ leads to a reduction in the already negatively biased $O_3$ from the model, with the mean bias changing from -7.3 to -8.6 µg m$^{-3}$ (-11% to -13%). Similarly, simulated $O_3$ concentrations downwind of the city (in analogue to the downwind $NO_2$ concentrations

described above) are biased negatively in both base run and sensitivity study in July. The bias of downwind concentrations





changes from -5.4 µg m$^{-3}$ (-7%) in the base run to -6.8 µg m$^{-3}$ (-9%) in the sensitivity run. The negative bias in both NO$_x$ and O$_3$ in the base run is consistent with the model simulating insufficient NO$_x$ emissions in a NO$_x$-limited ozone production regime. The reduction of O$_3$ concentrations in response to increased NO$_x$ emissions is however consistent with the model actually being in a NO$_x$-saturated (VOC limited) ozone production regime. The representation of VOC emissions in the model

could play a role in explaining this discrepancy, as for example biogenic VOC emissions in the Berlin-Brandenburg urban area are underestimated when using WRF-Chem and MEGAN (Churkina et al., 2017). A comprehensive analysis of the simulated ozone production regime is beyond the scope of this work.

## 6.3 Analysis based on traffic counts

The model bias and the calculated correction factors show a diurnal cycle, with a larger model bias/correction factor in the
morning hours. As explained in Sect. 5.3 and 5.4, one reason for this might be differences between prescribed and real-world diurnal cycles of the emissions. The diurnal cycle of traffic emissions in the model is calculated based on traffic counts for Berlin, assuming a linear scaling of traffic emissions with traffic counts, as done in many modelling studies.

Here, we use three years of hourly observations of roadside NO$_x$ concentrations and traffic counts measured at the same stations in order to get insights into the relationship between NO$_x$ concentrations and traffic counts. A linear regression
model does not explain the variance of observed NO$_x$ concentrations at nighttime, as indicated by the R$^2$ close to 0 in Fig. 9. However, during daytime, traffic counts alone explain up to ca. 40% of observed NO$_x$ variance, particularly during the traffic rush hours. The explained variance is smaller during the afternoon peak. In comparison to a linear model, a quadratic model (NO$_x$ ∝ (traffic_count)$^2$) does not explain more of the observed variance (not shown). An exponential model (NO$_x$ ∝ exp(traffic_count)), however, does explain a considerably larger share of the observed variance during daytime and particularly
during the traffic rush hours, as depicted in Fig. 9 (up to ca. 60% depending on the station).

This simple comparison suggests that roadside NO$_x$ concentrations, and thus most likely also road transport NO$_x$ emissions, scale more than linearly with traffic counts at times when the traffic intensity is high and underline that the assumption of a linear scaling of traffic emissions with traffic counts does not reflect the diurnal variation of traffic emissions sufficiently. More highly congested roads are typical in the morning, and emission factors are higher in congested situations compared
to free flowing traffic. Differences in congestion could contribute to explaining the non-linear scaling of NO$_x$ concentrations with traffic intensity. While the impact might not be large when simulating air quality with coarser models, it might play a more important role for high resolution air quality modelling, and the temporal distribution of emissions could potentially be improved when taking these differences into account.

## 6.4 Discussion of traffic emissions

Based on a comparison of modelled with observed NO$_2$ concentrations, we estimate that traffic emissions in the urban core of Berlin are underestimated by a factor of ca. 3 on weekdays between ca. 6:00 and 17:00 UTC. This corresponds to an overall underestimation of NO$_x$ traffic emissions (all day average) in the urban centre by a factor of ca. 2, and an underestimation of total NO$_x$ emissions (all day average) in the urban centre by a factor of ca. 1.5. Reasons for the underestimation of emissions



used in this study can include limitations in the applicability of the emission inventory used here for high resolution urban air quality modelling, problems in the temporal distribution of emissions, but also a general underestimation of traffic $NO_x$ emissions in the inventories. These three points are discussed further in the following.

First, while a reasonably good model performance can be achieved using the downscaled version of the TNO-MACC III
inventory outside of the urban areas, the deviations of modelled from observed $NO_2$ in the urban background might point to limitations in the applicability of these types of emission inventories for high resolution modelling of $NO_2$ in urban areas. The horizontal resolution of the original TNO-MACC III emission data is ca. 7 km x 7 km and national totals are disaggregated on the grid based on traffic intensities. Spatial differences in congestion, with emissions greatly varying between the different driving conditions and with car speed (e.g. Hausberger and Matzer, 2017), are probably not well resolved. A comparison of
the downscaled version of the TNO-MACC III inventory for Berlin with a local inventory has, however, not revealed major differences in road transport emissions (see Sect. 2.4.3), suggesting that a static highly resolved local inventory based on detailed local information is not likely to improve the model results by much.

Second, in addition to spatially unresolved differences in driving conditions and related emission factors locally increasing the underestimation of emissions, the diurnal cycle of the bias in all seasons suggest that the diurnal cycle of traffic emissions
also does not sufficiently account for temporal differences in driving conditions. This is consistent with the observation-based analysis, suggesting that observed $NO_x$ concentrations do not scale linearly with traffic counts. While these assumptions might be valid for coarser model resolutions, they may need to be revisited when going to higher resolutions with a focus on urban areas. However, modelled $NO_2$ concentrations are broadly underestimated throughout the day, which means that deviations of the model diurnal cycle from the real-world diurnal cycle alone cannot explain the underestimation of modelled $NO_2$ and $NO_x$
concentrations.

Third, traffic $NO_x$ emissions may be underestimated generally by emission inventories. The correction factor calculated here is in line with the results from other studies quantifying traffic emission underestimations in Europe, reporting traffic $NO_x$ underestimations of around 80% (Lee et al., 2015), a factor of 1.5-2 (Lee et al., 2015) and up to a factor of 4 (Karl et al., 2017). A potential reason for the underestimation in $NO_x$ emissions from traffic can be discrepancies between real-world emission
factors and those used in emission inventories. Even though HBEFA emission factors, which are often used for calculating emissions, are based on real-world driving conditions, the latest update of the handbook reports higher emission factors than previously assumed for Euro 6 and Euro 4 diesel cars (Hausberger and Matzer, 2017), e.g. an increase by ca. 50% in case of Euro 6 vehicles (Fig. 14 in Hausberger and Matzer, 2017). In addition, the update assesses the temperature dependence of emission factors and concludes that it may lead to increases in $NO_x$ emissions of more than 30%, compared with standard test
conditions. $NO_x$ emissions from diesel cars increase with decreasing temperatures (Hausberger and Matzer, 2017). This may also contribute to the larger correction factor calculated for the winter months. Finally, while some amount of congestion is included/assumed in the emission inventories, this might be an aspect that is underestimated in terms of severity and extent.

The first and second point of this discussion suggest that improvements might be achieved by combining high resolution chemical transport models with more detailed approaches of calculating emissions. Coupling with a traffic model, for example,
might allow for not only being able to take local difference in traffic conditions into account, but also prescribe a more realistic



diurnal cycle of traffic emissions. Dispersion modelling and street canyon modelling (e.g. OSPM, Berkowicz, 2000) often already take a more detailed calculation of traffic emissions into account, and different emission modelling approaches exist (e.g. traffic models such as MATSim, Horni et al., 2016). The benefit of high resolution chemistry transport modelling, e.g. their ability to assess the impact of different emission sources on air quality on larger scales and downwind of the main emission

sources, could be further exploited if coupled with existing, more detailed approaches in calculating traffic emissions or general improvements in the accuracy and resolution of emission inventories.

    The consistent findings that inventories of European traffic emissions may be underestimated, coming from studies using very different methodologies, suggests that further research is necessary in order to understand real-world traffic emissions and represent them in the inventories accordingly. Alternative measurement approaches could help verify the assumptions

underlying the calculation of emissions, and help identify potential systematic problems.

## 7    Summary and conclusions

Several modelling studies, particularly for Europe, have reported an underestimation of modelled $NO_2$ concentrations compared with observations. Measurement studies also suggest that there might be considerable differences between measured urban $NO_x$ emissions and emissions provided by emission inventories based on official reporting, particularly when the contri-

bution of traffic is large. This study quantifies the underestimation of traffic $NO_x$ emissions using WRF-Chem in a top-down approach, with the Berlin-Brandenburg area in Germany as a case study. The emission inventory used here is TNO-MACC III, downscaled to 1 km x 1 km over the Berlin-Brandenburg area based on local proxy data. The downscaled traffic emissions averaged over Berlin only differ by 6% from a local bottom-up traffic emission inventory.

    A diagnostic evaluation of the model results shows that particularly in the rural and suburban background, the long term

and synoptic components representing processes at time scales of the order of 2.5 to 21 (synoptic) and longer than 21 days (long term) are simulated well by the model. This suggests that the modelled impact of meteorology on concentrations is represented well overall. The largest contribution to the model error comes from the (negative) bias in the urban background, and from deviations of modelled from observed variability of the diurnal component (0.2-2.5 days). This suggests a possible underestimation of urban emissions, of which traffic is the single most important contributor to $NO_x$ emissions, but is also

consistent with deficiencies in other processes varying on the diurnal scale such as the modelled mixing in the planetary boundary layer. The analysis of the model results suggests that the latter is particularly relevant in summer and spring, and that further research is needed in order to better represent urban processes in WRF-Chem. For example, the parameterization of urban processes needs to better account for urban heat and momentum fluxes for a more realistic representation of mixing both at daytime and at nighttime, particularly in summer. In addition, measurements of vertical profiles of $NO_x$ in urban areas are

needed to evaluate and improve models for applications in urban areas.

    The analysis of the diurnal cycle of the model bias as well as a simple observation-based calculation showing that roadside $NO_x$ concentrations scale non-linearly with traffic counts suggest that a further source of error is likely the prescribed diurnal cycle used for traffic emissions. In this study as well as in many other modelling studies, the diurnal cycle of traffic emissions





is calculated assuming a linear scaling of traffic emissions with traffic counts. While this might be sufficient for coarser model resolutions, high resolution urban air quality modelling with chemistry transport models might benefit from a more detailed temporal distribution not only taking into account traffic intensity via a scaling with traffic counts, but also diurnal differences in congestion.

We quantify the underestimation of traffic emissions based on the finding that the weekday bias in modelled $NO_2$ is larger than on weekends and that the contribution of traffic $NO_x$ to total $NO_x$ emissions in the urban area is typically higher on weekdays. The results suggest that traffic emissions are underestimated by ca. a factor of 3 in the core urban area on weekdays when traffic is highest (6:00 to 17:00 UTC). The underlying assumption is that other sources of model errors influence the model bias equally on weekdays and weekends, with the underestimation of traffic emissions having the largest effect on
modelled $NO_2$ concentrations on weekdays. This underestimation corresponds to an underestimation of weekly mean traffic $NO_x$ emissions in the core urban area of ca. a factor of ca. 2 and an underestimation of total $NO_x$ emissions in the city centre by a factor of ca. 1.5. Two sensitivity simulations for January and July 2014 with $NO_x$ emissions from traffic scaled with the estimated correction factor show that increased traffic emissions improve the model bias in $NO_2$ and $NO_x$ concentrations in both seasons in the urban background, and also improved modelled downwind concentrations. The still negative bias is
consistent with the factor being a rather conservative estimate.

The underestimation of $NO_2$ concentrations throughout the day, the consistency of the calculated correction factor with findings from other studies and the improvement of model results applying the correction factor suggest that more research is needed in order to more accurately understand the spatial and temporal variability in real-world $NO_x$ emissions from traffic, and apply this understanding to the inventories used in high resolution chemical transport models.

**8    Code availability**

WRF-Chem is an open-source, publicly available community model. A new, improved version is released approximately twice a year. The WRF-Chem code is available at http://www2.mmm.ucar.edu/wrf/users/download/get_source.html. The corresponding author will provide the modifications introduced and described in Sect. 2.1 upon request.

**9    Data availability**

The observational and model input data used in this study are publicly available (references indicated in the manuscript) or available upon request.

*Acknowledgements.* This work was hosted by IASS Potsdam, with financial support provided by the Federal Ministry of Education and Research of Germany (fBMBF) and the Ministry for Science, Research and Culture of the State of Brandenburg (MWFK). The authors would like to thank Hugo Denier van der Gon and Jeroen Kuenen (TNO) for providing data of and information pertaining to the TNO-MACC III
inventory and cooperation on the downscaling of the data to a higher resolution; Klaus Schäfer (KIT) for ceilometer data; and Mark Lawrence



(IASS), Martijn Schaap (TNO), Ravan Ahmadov (NOAA), Joana Leitao (IASS), Stefano Galmarini (JRC) and Betsy Weatherhead (NCAR) for discussions that helped shape the manuscript.





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

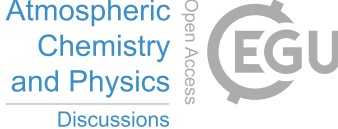

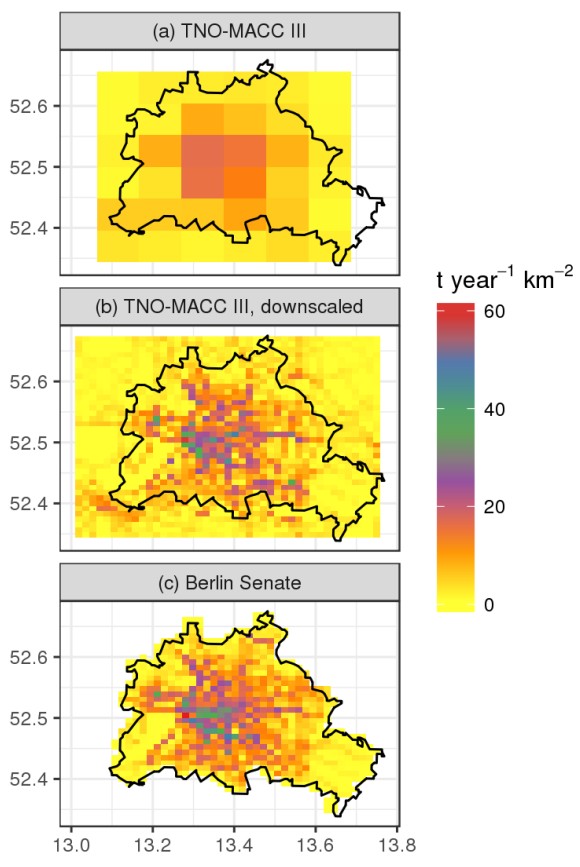

**Figure 1.** Total NO$_x$ emissions from traffic in Berlin; (a) from the TNO-MACC III inventory, 2011; (b) from the TNO-MACC III inventory, downscaled to a horizontal resolution of ca. 1 km, 2011; (c) from the Berlin Senate Department for the Environment, 2009.



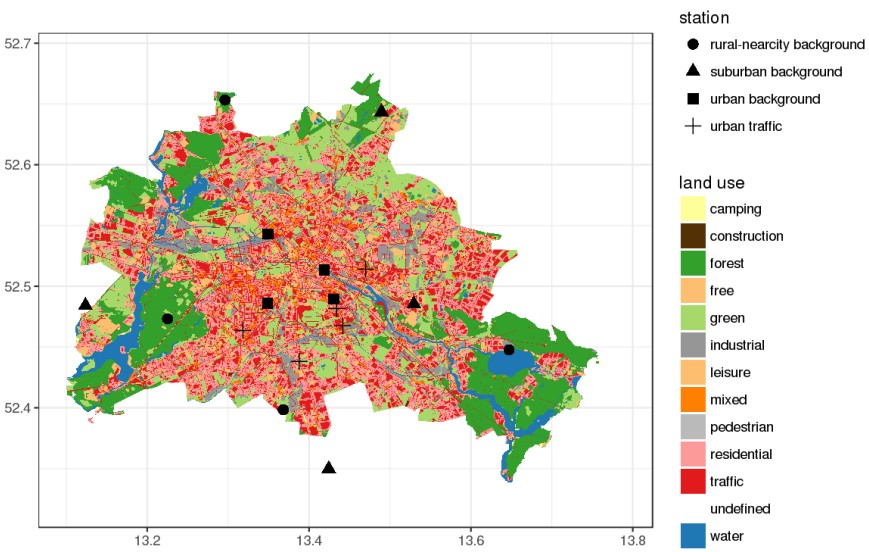

**Figure 2.** Locations of measurement stations in and close to Berlin, including their AirBase station area classification and type and the land use classes in Berlin according to Berlin Senate Department for Urban Development and Housing (the 2015).





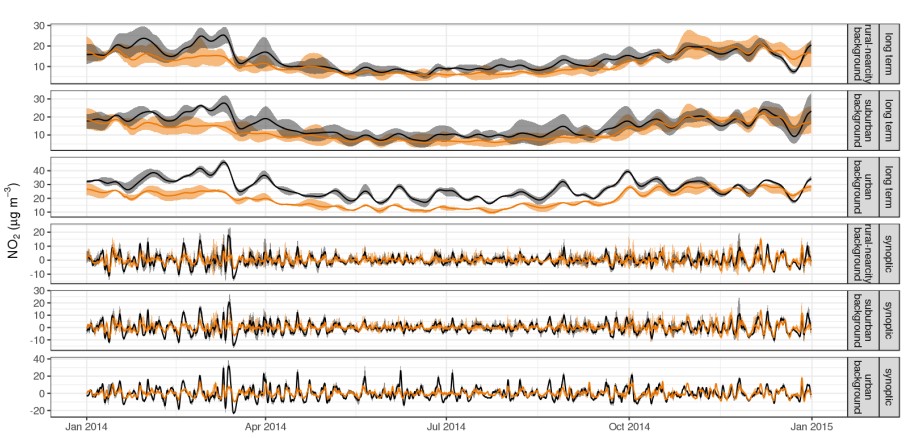

**Figure 3.** Long term and synoptic components of modeled (orange) and observed (black) time series, averaged over all stations of each station class. The shaded areas show the variability (25th and 75th percentiles) between the different stations within each class. Note the variable y-axis.





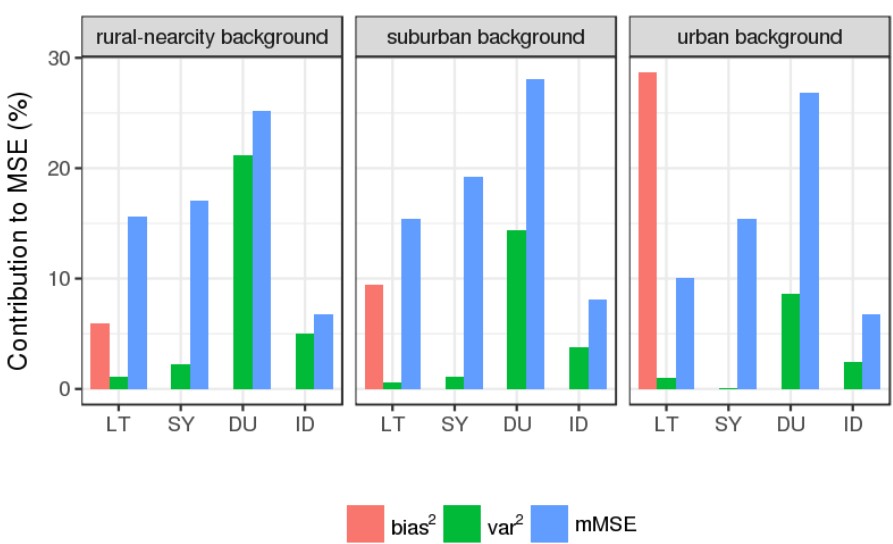

**Figure 4.** Contribution of different types of error to the mean square error of the model, per station class. The mean square error is divided into squared bias (bias$^2$), variance error (var$^2$) and minimum mean square error (mMSE) of the long term (LT), synoptic (SY), diurnal (DU) and intra-diurnal (ID) components (see Sect. 4.1 for further details).




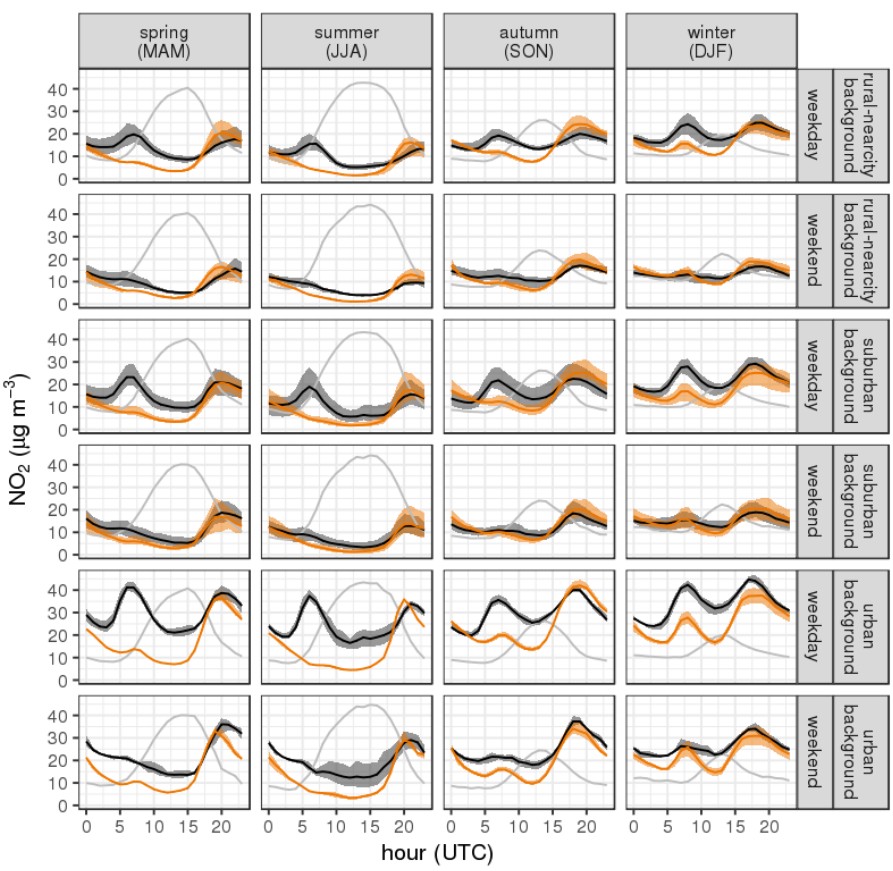

**Figure 5.** Mean diurnal cycles of modelled (orange) and observed (black) NO$_2$ concentrations, by station class and weekday/weekend. Shaded areas show the variability between the different stations' mean diurnal cycles (25th and 75th percentiles). Grey lines show the mean modeled planetary boundary layer heights at the respective grid points (scaled, but the relative changes between different hours and seasons are maintained).





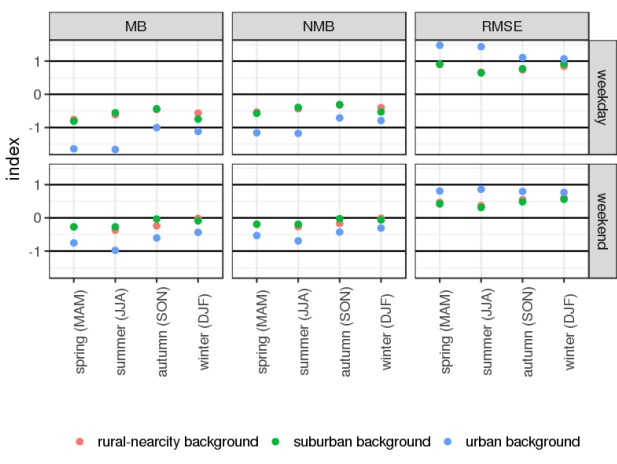

**Figure 6.** Skill of WRF-Chem in simulating daytime (6-17 UTC) observed $NO_2$ concentrations. The index represents the the model quality objective for the root mean square error (Sect. 4.1) and the performance criteria for mean and normalized mean bias (described in the Supplementary material), for weekend and weekday days and each month/season.





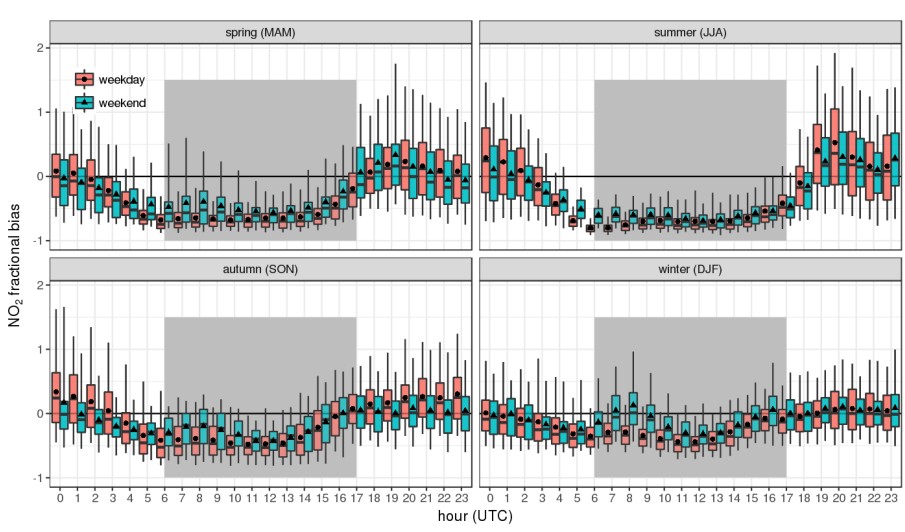

**Figure 7.** Relative bias in modeled NO$_2$ concentrations at urban background sites in Berlin, averaged over each season, hour and weekend/weekday. The boxplot show median (line), 25th and 75th (box), 5th and 95th (whiskers) percentiles of the hourly bias. Points show the mean. The grey shaded area shows the time period considered for quantifying the underestimation of daytime traffic emissions.





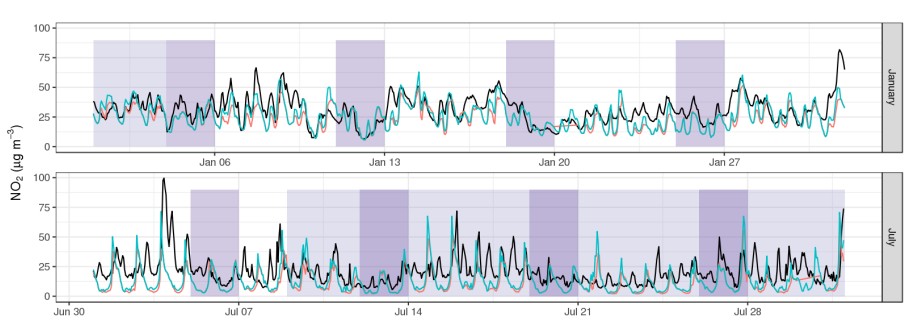

**Figure 8.** Time series of hourly observed (black line) and modelled $NO_2$, comparing the base simulation (red) with the sensitivity simulations using increased traffic emissions by a factor of 3 between 6:00 UTC and 17:00 UTC on weekdays. The time series are averaged over all 4 urban background stations. Weekends are highlighted in dark purple, and holidays are highlighted in light purple.





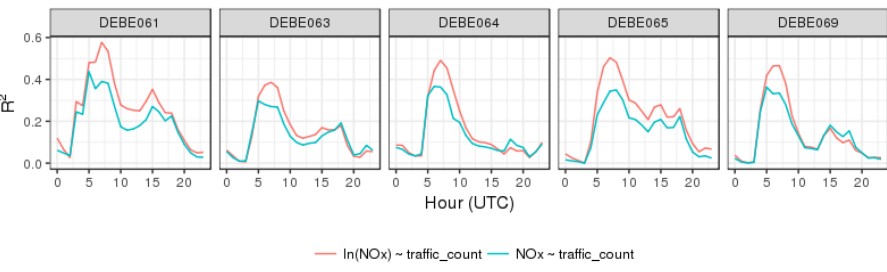

**Figure 9.** Comparison of $R^2$ for linear and exponential fit of roadside $NO_x$ concentrations with traffic counts.



**Table 1.** Model configuration and input data.

| Process | Option/dataset | Remarks |
|---|:---:|---:|
| Land surface model | Noah LSM | CORINE land use data |
| Urban processes | single layer UCM | 3 categories: roofs, walls, trees |
| Boundary layer | MYNN | |
| Cumulus convection | Grell-Freitas | switched on for both domains |
| Cloud microphysics | Morrison double-moment | |
| Radiation (sw+lw) | RRTMG | |
| Aerosols | MADE/SORGAM | chem_opt=106 |
| Chemistry | RADM2 | with KPP |
| Photolysis | Madronich F-TUV | |
| Anthropogenic emissions | TNO-MACC III | see Sect. 2.4 for details |
| Biogenic emissions | online | MEGAN |
| Dust and sea salt emissions | online | dust_opt=3, seas_opt=2 |
| Meteorological boundary conditions | ERA-Interim | sst_update=1 |
| Chemical boundary conditions | MOZART4-GEOS5 | |





**Table 2.** Model simulations presented in this paper.

| Simulation | Chemistry | Period | Emissions |
|---|---|:---:|---:|
| 2014_ref | RADM2 | 1.1.2014 - 31.12.2014 | TNO-MACC III |
| 2014_moz | MOZART | July 2014 | TNO-MACC III |
| 2014_emis | RADM2 | January 2014 | TNO-MACC III |
| | | July 2014 | Traffic $NO_x$ increased (Sect. 6.1) |





**Table 3.** Modeled meteorology compared to observations, annual and seasonal performance indicators. Mean bias (MB) and root mean square error (RMSE) are indicated in µg m$^{-3}$, the normalized mean bias (NMB) and correlation coefficient (R) are unitless.

| | | Temperature | | | | Wind speed | | | |
|---|---|---|---|---|---|---|---|---|---|
| | | MB | NMB | RMSE | R | MB | NMB | RMSE | R |
| Lindenberg | 2014 | -0.66 | -0.06 | 2.17 | 0.96 | 0.81 | 0.25 | 1.63 | 0.69 |
| | spring (MAM) | -0.48 | -0.04 | 2.26 | 0.91 | 0.87 | 0.27 | 1.8 | 0.68 |
| | summer (JJA) | -0.89 | -0.05 | 2.4 | 0.89 | 0.87 | 0.31 | 1.68 | 0.61 |
| | autumn (SON) | -0.55 | -0.05 | 1.99 | 0.95 | 0.92 | 0.32 | 1.58 | 0.63 |
| | winter (DJF) | -0.7 | -0.3 | 2 | 0.93 | 0.57 | 0.14 | 1.43 | 0.76 |
| Potsdam | 2014 | -0.71 | -0.06 | 2.25 | 0.96 | -0.47 | -0.12 | 1.4 | 0.69 |
| | spring (MAM) | -0.49 | -0.04 | 2.16 | 0.93 | -0.34 | -0.08 | 1.42 | 0.73 |
| | summer (JJA) | -0.45 | -0.02 | 2.1 | 0.91 | -0.14 | -0.04 | 1.41 | 0.55 |
| | autumn (SON) | -0.76 | -0.07 | 2.34 | 0.94 | -0.46 | -0.12 | 1.27 | 0.62 |
| | winter (DJF) | -1.15 | -0.44 | 2.42 | 0.9 | -0.99 | -0.2 | 1.47 | 0.79 |
| Schoenefeld | 2014 | -0.61 | -0.06 | 2.16 | 0.96 | 0.02 | 0 | 1.3 | 0.78 |
| | spring (MAM) | -0.12 | -0.01 | 1.97 | 0.94 | 0.07 | 0.02 | 1.34 | 0.81 |
| | summer (JJA) | -0.63 | -0.03 | 2.1 | 0.92 | 0.16 | 0.05 | 1.39 | 0.66 |
| | autumn (SON) | -0.73 | -0.06 | 2.27 | 0.94 | 0.12 | 0.04 | 1.21 | 0.7 |
| | winter (DJF) | -0.98 | -0.39 | 2.3 | 0.91 | -0.31 | -0.07 | 1.25 | 0.85 |
| Tegel | 2014 | -1.25 | -0.11 | 2.48 | 0.96 | 0.4 | 0.12 | 1.33 | 0.75 |
| | spring (MAM) | -0.83 | -0.07 | 2.26 | 0.93 | 0.58 | 0.18 | 1.43 | 0.77 |
| | summer (JJA) | -1.02 | -0.05 | 2.23 | 0.92 | 0.58 | 0.2 | 1.44 | 0.66 |
| | autumn (SON) | -1.44 | -0.12 | 2.65 | 0.94 | 0.47 | 0.16 | 1.24 | 0.69 |
| | winter (DJF) | -1.72 | -0.54 | 2.76 | 0.9 | -0.08 | -0.02 | 1.17 | 0.84 |
| Tempelhof | 2014 | -1.21 | -0.11 | 2.51 | 0.96 | 0.45 | 0.13 | 1.32 | 0.74 |
| | spring (MAM) | -0.67 | -0.06 | 2.22 | 0.93 | 0.47 | 0.13 | 1.38 | 0.77 |
| | summer (JJA) | -1.17 | -0.06 | 2.35 | 0.91 | 0.5 | 0.16 | 1.51 | 0.63 |
| | autumn (SON) | -1.38 | -0.11 | 2.65 | 0.93 | 0.43 | 0.14 | 1.25 | 0.7 |
| | winter (DJF) | -1.63 | -0.54 | 2.76 | 0.9 | 0.38 | 0.1 | 1.12 | 0.81 |



**Table 4.** Modeled chemistry, seasonal performance indicators (averaged for each station class, each class includes 4 stations) and the model quality objective for NO$_2$. Mean bias (MB) and root mean square error (RMSE) are indicated in µg m$^{-3}$, the normalized mean bias (NMB) and correlation coefficient (R) are unitless.

| | | NO$_2$ | | | | NO$_x$ | | | | O$_3$ | | | | NO$_2$ |
| --- | --- | --- | --- | --- | --- | --- | --- | --- | --- | --- | --- | --- | --- | --- |
| | | MB | NMB | RMSE | R | MB | NMB | RMSE | R | MB | NMB | RMSE | R | MQO |
| rural- | 2014 | -2.12 | -0.16 | 10.2 | 0.5 | -3.77 | -0.23 | 15 | 0.48 | 5.02 | 0.11 | 22.49 | 0.7 | 0.78 |
| nearcity | autumn (SON) | -0.97 | -0.06 | 9.97 | 0.48 | -3.9 | -0.19 | 16.2 | 0.45 | 11.96 | 0.41 | 22.71 | 0.66 | 0.76 |
| backgr. | spring (MAM) | -2.91 | -0.23 | 11.69 | 0.42 | -4.26 | -0.29 | 16.39 | 0.37 | 3.88 | 0.07 | 23.42 | 0.62 | 0.89 |
| | summer (JJA) | -2.38 | -0.26 | 8.23 | 0.37 | -2.88 | -0.28 | 9.57 | 0.32 | 1.41 | 0.02 | 25.49 | 0.61 | 0.64 |
| | winter (DJF) | -2.2 | -0.12 | 10.66 | 0.47 | -4.08 | -0.18 | 16.83 | 0.46 | 2.85 | 0.09 | 17.18 | 0.55 | 0.79 |
| suburban | 2014 | -2.8 | -0.19 | 10.67 | 0.55 | -7.2 | -0.35 | 20.13 | 0.48 | 4.88 | 0.11 | 22.45 | 0.7 | 0.8 |
| backgr. | autumn (SON) | -0.76 | -0.05 | 10.32 | 0.52 | -7.92 | -0.32 | 23.12 | 0.44 | 12.22 | 0.42 | 22.39 | 0.67 | 0.78 |
| | spring (MAM) | -4.41 | -0.31 | 12.2 | 0.49 | -8.25 | -0.44 | 21.71 | 0.39 | 4.15 | 0.07 | 24.06 | 0.61 | 0.92 |
| | summer (JJA) | -2.88 | -0.29 | 9.01 | 0.44 | -5.12 | -0.4 | 13.14 | 0.34 | 1.16 | 0.02 | 25.49 | 0.64 | 0.7 |
| | winter (DJF) | -3.14 | -0.16 | 10.96 | 0.53 | -7.57 | -0.28 | 21.24 | 0.49 | 2.02 | 0.06 | 16.53 | 0.57 | 0.8 |
| urban | 2014 | -7.83 | -0.29 | 16.69 | 0.51 | -15.84 | -0.4 | 35.57 | 0.47 | 3.25 | 0.08 | 21.01 | 0.73 | 1.13 |
| backgr. | autumn (SON) | -4.89 | -0.17 | 13.9 | 0.55 | -16.9 | -0.36 | 37.3 | 0.48 | 9.09 | 0.37 | 19.69 | 0.71 | 0.95 |
| | spring (MAM) | -10.23 | -0.38 | 19.71 | 0.51 | -17.09 | -0.47 | 40.68 | 0.4 | 3.07 | 0.06 | 22.62 | 0.62 | 1.32 |
| | summer (JJA) | -9.26 | -0.41 | 18.16 | 0.36 | -13.3 | -0.47 | 28.92 | 0.24 | -1.94 | -0.03 | 24.85 | 0.6 | 1.28 |
| | winter (DJF) | -6.84 | -0.22 | 14.05 | 0.53 | -16.16 | -0.34 | 34.41 | 0.5 | 2.93 | 0.12 | 15.35 | 0.58 | 0.93 |





**Table 5.** Statistics of modelled $NO_2$ and $NO_x$ concentrations for January and July, for the base simulation and for the sensitivity simulation with increased traffic emissions, at the urban background stations in Berlin. Mean bias (MB) and root mean square error (RMSE) are indicated in µg m$^{-3}$, the normalized mean bias (NMB) and correlation coefficient (R) are unitless.

| | | NO₂ | | | | NOₓ | | | |
|---|---|---|---|---|---|---|---|---|---|
| | | MB | NMB | RMSE | R | MB | NMB | RMSE | R |
| Amrumer Str. | | | | | | | | | |
| Jan | 2014_ref | -6.77 | -0.21 | 11.93 | 0.65 | -14.63 | -0.31 | 28.85 | 0.65 |
| | 2014_emis | -4.16 | -0.13 | 11.09 | 0.68 | -8.01 | -0.17 | 26.18 | 0.64 |
| July | 2014_ref | -10.12 | -0.47 | 17.88 | 0.34 | -12.25 | -0.49 | 22.88 | 0.27 |
| | 2014_emis | -8.9 | -0.42 | 18 | 0.33 | -10.51 | -0.42 | 23.21 | 0.25 |
| Belziger Str. | | | | | | | | | |
| Jan | 2014_ref | -10.64 | -0.32 | 15.39 | 0.51 | -20.88 | -0.41 | 34.23 | 0.51 |
| | 2014_emis | -7.97 | -0.24 | 14.04 | 0.53 | -14.57 | -0.29 | 31.6 | 0.51 |
| July | 2014_ref | -7.32 | -0.37 | 15.32 | 0.31 | -8.38 | -0.37 | 17.81 | 0.23 |
| | 2014_emis | -4.02 | -0.2 | 16.73 | 0.22 | -3.67 | -0.16 | 20.43 | 0.12 |
| Nansenstr. | | | | | | | | | |
| Jan | 2014_ref | -6.09 | -0.21 | 11.12 | 0.61 | -12.81 | -0.3 | 25.05 | 0.56 |
| | 2014_emis | -3.72 | -0.13 | 10.28 | 0.64 | -7.44 | -0.18 | 23.28 | 0.55 |
| July | 2014_ref | -8.73 | -0.43 | 15.33 | 0.42 | -10.89 | -0.45 | 19.18 | 0.35 |
| | 2014_emis | -6.88 | -0.34 | 15.43 | 0.36 | -8.25 | -0.34 | 19.37 | 0.28 |
| Brückenstr. | | | | | | | | | |
| Jan | 2014_ref | -7.1 | -0.23 | 12.51 | 0.57 | -17.27 | -0.36 | 36.4 | 0.51 |
| | 2014_emis | -4.5 | -0.15 | 11.13 | 0.63 | -11.16 | -0.24 | 32.58 | 0.54 |
| July | 2014_ref | -9.92 | -0.46 | 17.24 | 0.25 | -12.85 | -0.49 | 22.7 | 0.17 |
| | 2014_emis | -8.05 | -0.37 | 16.87 | 0.26 | -10.16 | -0.39 | 22.13 | 0.18 |