# Peer review of "Top-down quantification of $NO_x$ emissions from traffic in an urban area using a high resolution regional atmospheric chemistry model"

_Atmospheric Chemistry and Physics, 2017_

## Referee Comment (RC1) · Anonymous Referee #1 · 20 Dec 2017

This paper investigates the possible causes of NOx model-measurement mismatch for high-resolution regional modelling. The authors use WRF-Chem, run for 1 year (through 2014) at high (3x3km) resolution, with TNO emissions downscaled using local data to improve the resolution of emissions. They use spectral decomposition of the modelled and observed timeseries to extract the contributions of processes across a range of timescales to the NOx system, finding that the model is good at capturing the long term and synoptic behaviour of the system, but struggles to capture variations at diurnal and intra-diurnal timescales. NO2 measurements at urban and suburban background measurement sites are underestimated (as seen in previous studies too), and they suggest that difficulties in capturing the behaviour of vertical mixing and traffic

emissions are most likely to cause the discrepancies between the model and measurements. The sensitivity of simulated concentrations to a top-down correction factor for NOx emissions using traffic data is investigated - based on this analysis they suggest that traffic emissions should be scaled non-linearly (as opposed to the current linear scaling) with traffic counts.

The paper is well written, and the study carried out in a careful and methodic manner. The difficulties of trying to compare ground-level, single point measurements with model predictions averaged over a (much) larger grid cell do limit the analysis that can be carried out in such a study, but I think the authors recognise and work within those limitations well, and the insights gained from this study will be relevent to the ACP readership. I recommend publication, subject to a few minor corrections.

Specific Comments:

1) Comparison of the modelled NO2 concentrations in this study with those in Kuik et al (2016) does suggest that the modification to the chemical mixing routines in WRF-Chem in order to set a minimum vertical mixing value helps reduce the overprediction of NO2 at nighttime. I am, however, uncomfortable with the some of the choices made in this routine: using fixed minimum mixing rates, and emissions as a proxy for land use, both seem like clunky fudges to me. CO from shipping emissions would lead to it being activated over the oceans, for example, which I think is counter to the intended influence of this modification. It is good that the authors have noted their use of this modification in this study (and disappointing that if it has been used in previous published studies, that it does not seem to have been documented as being used), as this will add to growing evidence in the literature (c.f. Hu et al, 2013) that this issue exists and needs addressing in a more rigorous and methodical manner.

As changes other than this modification have been made to the model setup between this study and Kuik et al (2016), it would be useful for readers if a figure illustrating the differences in NO, NO2, NOx and O3 concentrations in model simulations with and

without this modification could be added to the supplementary material (as this would be illustrative, it would only need to be for a couple of days, not for the entire year).

In addition, recent communication from Georg Grell on the wrf-chem-discussions list indicated that the ACM2 scheme now conducts mixing of chemical species within the PBL physics routines (improving consistency between the treatment of chemical and meteorological tracers). Though this is probably outside the scope of this study, have the authors considered investigating if this scheme improves nighttime pollutant concentrations?

2) The locations of measurement stations are not very clear in Figure 2, and it is currently impossible just using this paper to determine the exact location of individual named stations within the study. An extra map (either in the main paper, or in supplementary material), with the locations of named measurements stations made much clearer, would be very useful.

3) In Section 6.2 the authors use wind speed and direction data to select data for investigating the impact of changing urban traffic emissions on downwind model NOx predictions. Were there a specific reasons for chosing 2 m/s wind speed and 72 data pairs as your cut-off points? How dependent on your data selection criteria are the changes in model bias between simulations - if you adjusted the criteria by +/-10% would your results change greatly?

Technical Corrections:

1) It might be helpful to expand the abbreviations MAM, JJA, SON, and DJF when they first appear, to make the paper more accessible to those who are not familiar with these common abbreviations.

2) page 3, line 7: it would be more correct to use "and" instead of "with" in this sentence.

3) page 8, 2nd paragraph: urRV needs proper formatting twice here.

4) page 16, line 2: the correction factor should be "f", not "F"?

5) page 17, line 19: should it be "wind direction bin", instead of "wind speed bin"?

6) page 34, figure 8: your description misses out the colour (blue?) of the sensitivity simulation data

7) page 38, table 3: in the table caption, the units of MB and RMSE need correcting from ug/m3 to K and m/s (respectively).

References:

Hu, X.-M., P. M. Klein, and M. Xue (2013), Evaluation of the updated YSU planetary boundary layer scheme within WRF for wind resource and air quality assessments, J. Geophys. Res. Atmos., 118, 10,490–10,505, doi:10.1002/jgrd.50823.

---

## Referee Comment (RC2) · Anonymous Referee #2 · 11 Jan 2018

The paper analyses the general problem of simulating NOx concentrations in urban areas using high resolution chemical transport models, as in this case WRF-Chem. Via top-down quantification of NOx emissions from traffic the authors claim the need for a correction factor for traffic NOx and highlight the general model problems in simulating meaningful mixing in the urban boundary layer – which they state being a main source of error in modelling near surface concentrations of air pollutants. A detailed model evaluation is provided which is underpinned by a comprehensive statistical analysis. The paper is well written and structured in an understandable way.

In its current form, it shows a case study for the city of Berlin which maybe should

be included in the title. It would be nice if the authors could add a section or a short passage, discussing the questions What do we learn from this study? or, Are the results transferrable to other urban areas? In the following you can find comments line by line:

Page 1: Line 11: It is unclear at this point you have treated the discrepancy between model resolution and resolution of the emission data. What is the purpose of the emission downscaling to 1km and what was the initial resolution of the inventory. These points should be discussed more details in the methodology section, also explaining the downscaling method.

P2: 9ff: better EEA 10: What is the reason for the downward trend? 16: specify 'real world driving condition' 21: better WHO

P3: 3,7: Fallmann et al. 2016 presents a modelling study for the city of Stuttgart following similar procedures and gets similar findings 21: unclear: 'basing'

P4: 1: what is meant by 'activity data' 5: where does the large error range come from and how did you consider this in your study? 14: better 'model setup' 16: see comment above about the purpose of downscaling 21: State, whether the sensitivity simulations are taken from the output of the 1 year run or whether they are independent experiments. General: specify the source and the characteristics of the urban canopy parameters

P5: 20ff: how is second part of the simulation initialized, how does it refer to the first model period? 28: Discuss briefly if you can estimate an error from using 2011 emission data for a 2014 simulation?

P6: 2: specify chosen categories and reason for the selection 23: Can you give an estimate of the error in mean values when only considering weekdays?

P7: 25: Procedure of NO2 retrieval at Air Base station unclear. How do you get the final NO2 concentrations from converted NO?

P8: 20,21: Specify more details about 'using the model setup for [. . .] policy relevant'. Do you mean operational forecast here?

P10: 25: state briefly why you changed the model setup compared to Kuik et al (2016) with regard to re-initialization etc. Can you discuss the impact of the modifications on NO, NO2, NOx, O3? 30: reference height for wind speed and temperature? Model height 30m?

P11: 5-21: Indicate briefly the reason for different model performance for different seasons/daytime → does that relate to general problems with tour model setup (boundary layer scheme, chemical mechanism etc.)

5.1/5.2: how do the biases in meteorology relate to biases in chemical species? Is there a seasonal dependence?

P13: 24: How is the link between population density and emission achieved/applied in the model?

30/31: earlier it was mentioned that only weekday profiles were applied in the model setup. Please clarify this aspect here.

P14: 5ff: Does a change of the urban canopy model make a change in boundary layer mixing close to the surface? How was the change in model code achieved? What is the model height for the NOx evaluation? How do model and observation height compare?

Some boundary layer schemes have been modified recently in order to take into account the mixing of chemical compounds inside the boundary layer. Can you comment on this? What impact might this have to your study?

14: better 'too strong' instead of 'too efficient' 15: Did you find a sensitivity to UCM selection? 25: Is the emission at urban background station generally too low? Maybe you are missing an advection term here?

P15: 16: what are the dominant non-traffic emission sources in the area of Berlin in

summer time?

P16: 18: Are the 'newer' diesel cars already implemented in the inventory used here?

P17: 6: From your experience: Which part of the model/configuration has to be changed for improving the representation of mixing processes? 11: Can the difference between secondary and primary NOx be attributed to wrong NO titration processes? Or other chemical processes?

P18: 11: The fact that a higher resolution emission inventory does not improve the model results is an important finding here and has to be explained more detailed. Did you run model experiment with a high-resolution emission inventory as well? What is the resolution here?

15-20: What kind of models do you suggest in terms of higher resolution? Chemical transport models, LES, CFD, dispersion models...? 25: meaning: HBEFA

P20: 27: Urban processes in WRF-Chem are linked via meteorology only. Do you have a suggestion, how UCMs have to be changed in order to improve AQ simulations?

P21: Some open questions should be briefly discussed her as mentioned in the beginning:  c Why is the study important for the field of research?  c What were the novel findings?  c What can we learn?  c Are the results transferrable to other urban areas?

---

## Referee Comment (RC3) · Anonymous Referee #3 · 14 Jan 2018

The authors discuss the well-known problem of the underestimation of simulated daytime NO$_x$ concentrations in urban areas. WRF-Chem is applied at a 3 km x 3 km horizontal resolution for Berlin for the whole year 2014. The impact of a possible underestimation of traffic emissions is investigated. Spectral decomposition of observed and modelled time series and error apportionment suggests that an underestimation in traffic emissions is likely one of the main causes of the bias in the modelled NO$_2$ concentrations.

The paper is well written and can be accepted after minor revision. In addition to the reviewer comments 1 and 2 I just have to add some minor comments – in particular

related to the simulation setup.

Page 4, line 29: 'top' should be added after 'model layer'. With a lowest layer depth of 30 m the near surface profiles are not well resolved. How does this affect the simulated near-surface $NO_x$ concentrations?

Page 5, lines 1 – 4 and page 10, line 8: This fix of the too weak vertical exchange during nighttime seems quite arbitrary. Is this fix only applied only to pollutant concentrations? Enhanced nocturnal mixing would also affect the thermal stratification which could in turn affect the vertical exchange of pollutants. Therefore, the applied fix should be commented more critically.

Page 5, line 21: Why is a 4-day spin-up required for the simulation of the last 6 month of the year?

Page 6, line 9: Please mention also the heights and not also the layer numbers.

Page 6, line 23: Why is the same diurnal cycle applied for weekdays and weekends? The traffic counts show certainly a different course for weekdays and weekends.

Page 10, line 3: The linking between NO, $NO_2$ and $O_3$ is also true for offline models.

Page 10, line 25: Please add some details about the 'misreprentation of the diurnal cycles'.

Minor issues:

Page 6, line 28: Does the percentage refer to $NO_2$ mass? Please clarify.

Page 8, line 25: The equation for MQO should be inserted already here.

Page 13, line 9: This is the case for all simulations with a grid width of only 3 km.

Eea and Wmo should be capitalized

---

## Author Comment (AC1) · 15 Apr 2018

**Referee comments, answers and changes in the manuscript**

**General comments**

**Referee 1**

Comparison of the modelled NO2 concentrations in this study with those in Kuik et al (2016) does suggest that the modification to the chemical mixing routines in WRF- Chem in order to set a minimum vertical mixing value helps reduce the overprediction of NO2 at nighttime. I am, however, uncomfortable with the some of the choices made in this routine: using fixed minimum mixing rates, and emissions as a proxy for land use, both seem like clunky fudges to me. CO from shipping emissions would lead to it being activated over the oceans, for example, which I think is counter to the intended influence of this modification. It is good that the authors have noted their use of this modification in this study (and disappointing that if it has been used in previous published studies, that it does not seem to have been documented as being used), as this will add to growing evidence in the literature (c.f. Hu et al, 2013) that this issue exists and needs addressing in a more rigorous and methodical manner.

As changes other than this modification have been made to the model setup between this study and Kuik et al (2016), it would be useful for readers if a figure illustrating the differences in NO, NO2, NOx and O3 concentrations in model simulations with and without this modification could be added to the supplementary material (as this would be illustrative, it would only need to be for a couple of days, not for the entire year).

In addition, recent communication from Georg Grell on the wrf-chem-discussions list indicated that the ACM2 scheme now conducts mixing of chemical species within the PBL physics routines (improving consistency between the treatment of chemical and meteorological tracers). Though this is probably outside the scope of this study, have the authors considered investigating if this scheme improves nighttime pollutant concentrations?

**Referee 3**

Page 5, lines 1 - 4 and page 10, line 8: This fix of the too weak vertical exchange during nighttime seems quite arbitrary. Is this fix only applied only to pollutant concentrations? Enhanced nocturnal mixing would also affect the thermal stratification which could in turn affect the vertical exchange of pollutants. Therefore, the applied fix should be commented more critically.

**Answer**

We share the referees' concern on the modification of the chemical mixing routine. This was a very much discussed topic at a workshop of European WRF-Chem users in Germany last year. Several points can be commented on:

- The fix is only applied to pollutant concentrations.
- As the focus of this study is Berlin, which is sufficiently far away from the sea, we do not believe that a potential modification of mixing when shipping emissions are high would impact our results. However, we share the referee's concern. An "easy" (yet still clunky) fix to the shipping emissions problem would be to restrict the increased mixing to grid cells with an urban land use category.
- Illustrative figures have been added to the supplementary material comparing results

from two simulations, one with default TNO diurnal cycles of emissions and no enhanced mixing, and one with diurnal cycles for Berlin and enhanced mixing. In addition, a reference to the supplementary material has been added in the manuscript as indicated below.

- At the time of running the simulations and writing, we were not aware of the new features of the ACM2 scheme, so we did not try it (though we did compare the results from the YSU, MYNN and MYJ schemes). We included the existence of this option in the conclusions as indicated below, note that this could also be an option for a more consistent treatment of meteorology and chemistry over urban areas, and suggest testing it as a follow-up of this study.

**Changes in the manuscript:**

Page 5, line 4: "Nighttime mixing over urban areas is not accounted for sufficiently by the urban parametrization and the PBL scheme and thus adjusted (dry\_dep\_driver.F) as described in the Supplementary Material. There, we also show illustrative results (Fig. S1-S3) of two test simulations comparing the impact of changes in this model setup with respect to Kuik et al. (2016), including the modification of nighttime mixing and the modification of the diurnal distribution of emissions (see Sect. 2.4)."

Page 20, line 24: "[...] but is also consistent with deficiencies in other processes varying on the diurnal scale such as the modelled mixing in the planetary boundary layer. The analysis of the model results suggests that the latter is particularly relevant in summer and spring, and that further research is needed in order to better represent urban processes and their coupling with chemistry in WRF-Chem. For example, the changes in the model code applied here to improve nighttime mixing can be critically discussed, and would ideally be replaced by an improved the parameterization of urban processes. The latter would needs to better account for urban heat and momentum fluxes for a more realistic representation of mixing both at daytime and at nighttime, particularly in summer. An alternative model configuration to be tested could be the recently extended ACM2 planetary boundary parametrization (Pleim et al., 2007), which now conducts mixing of chemical species within the planetary boundary layer scheme. In addition, measurements of vertical profiles of NOx in urban areas are needed to evaluate and improve models for applications in urban areas."

**Referee 1**

The locations of measurement stations are not very clear in Figure 2, and it is currently impossible just using this paper to determine the exact location of individual named stations within the study. An extra map (either in the main paper, or in supplementary material), with the locations of named measurements stations made much clearer, would be very useful.

**Answer:**

Manuscript updated as suggested.

**Changes in the manuscript:**

To clarify this, a map with station types and airbase codes as well as a table matching the codes with the station names and coordinates has been included in the supplementary material. The material has also been referenced in the main text:

Page 7 lines 12-14: "For the comparison with model results, observations from stations

within Berlin and in the adjacent surroundings in the Federal State of Brandenburg representing "urban background", "suburban background" and "rural near-city" conditions are used (Fig. 2, also see supplementary Figure S4 and Table S1)"

**Referee 1:** In Section 6.2 the authors use wind speed and direction data to select data for investigating the impact of changing urban traffic emissions on downwind model NOx predictions. Were there specific reasons for choosing 2 m/s wind speed and 72 data pairs as your cut-off points? How dependent on your data selection criteria are the changes in model bias between simulations - if you adjusted the criteria by +/-10% would your results change greatly?

**Answer**

The reason for choosing a wind speed threshold was to exclude situations in which the "downwind" station would not be affected by emissions in the city center. Results were only considered for a certain number of data pairs in order to get a sufficiently large number of hourly observations for averaging the results. If the wind speed threshold is changed by +/-10%, the results are not changed greatly, and the ranges of bias reduction indicated in the manuscript are still valid for January, and slightly enlarged for July, with both lower and higher changes (see table included below). Most stations excluded by the threshold of available data pairs have far fewer data pairs available than the threshold, and most stations included have a considerably larger number of data pairs available. An exception are the results for the stations Müggelseedamm and Johanna and Willi Brauer Platz in January, with 64 and 69 available data pairs, both located in the east of the city center. For the latter, the same pattern as reported above is true (higher concentrations simulated in the sensitivity simulation) and is thus in line with the reported results; however the model bias becomes positive in the sensitivity simulation. The model bias for the station Müggelseedamm is unchanged (and positive in both cases), which suggests that the location might not lie within the plume from the city on the selected dates. This is plausible, as the wind direction bin is rather large (90°). Overall, we conclude that the chosen thresholds are suitable for assessing the effect on downwind concentrations.

Table 1: Sensitivity of changes in NO2 model bias depending on the chosen cut-off for wind speed and the number of data pairs to be included. "n" denotes the number of data pairs, "MB" the mean bias averaged over all available data pairs in the respective wind direction bin (see main text). 1.8, 2.0 and 2.2 refer to the cut-off wind speed used for the calculations (in m/s). "diff" indicated the difference in mean bias between the base run and the sensitivity simulation. Highlighted are those stations excluded from the analysis, as the number of available data pairs is below the threshold (72).

| name                    | mo
nth | simula
tion  | n_
1.8 | MB_
1.8 | n_
2.0 | MB_
2.0 | n_
2.2 | MB_
2.2 | diff_
1.8 | diff_
2.0 | diff_
2.2 |
|-------------------------|-----------|-----------------|-----------|------------|-----------|------------|-----------|------------|--------------|--------------|--------------|
| Blankenfeld
e-Mahlow | Jan       | base            | 32        | 1.48       | 32        | 1.48       | 32        | 1.48       | 0.01         | 0.01         | 0.01         |
| Blankenfeld
e-Mahlow | Jan       | sensitiv
ity | 32        | 1.47       | 32        | 1.47       | 32        | 1.47       |              |              |              |
| Buch                    | Jan       | base            | 208       | -1.68      | 203       | -1.98      | 199       | -2.31      | -2.07        | -2.11        | -2.13        |

| Buch                                                                                                                                                                                                                             | Jan                                                                | sensitiv                                                                                                                                                       | 209                                                                            | 0.38                                                                                                            | 204                                                                                  | 0.13                                                                                                             | 200                                                                                        | -0.18                                                                                                                    |                                                    |                                                    |                                                    |
|----------------------------------------------------------------------------------------------------------------------------------------------------------------------------------------------------------------------------------|--------------------------------------------------------------------|----------------------------------------------------------------------------------------------------------------------------------------------------------------|--------------------------------------------------------------------------------|-----------------------------------------------------------------------------------------------------------------|--------------------------------------------------------------------------------------|------------------------------------------------------------------------------------------------------------------|--------------------------------------------------------------------------------------------|--------------------------------------------------------------------------------------------------------------------------|----------------------------------------------------|----------------------------------------------------|----------------------------------------------------|
| Frahnou                                                                                                                                                                                                                          | lon                                                                | ity                                                                                                                                                            | 101                                                                            | 2.02                                                                                                            | 100                                                                                  | 2.10                                                                                                             | 170                                                                                        | 2 22                                                                                                                     | 1 50                                               | 1.50                                               | 1 55                                               |
| Fronnau                                                                                                                                                                                                                          | Jan                                                                | Dase                                                                                                                                                           | 104                                                                            | -3.03                                                                                                           | 102                                                                                  | -3.19                                                                                                            | 179                                                                                        | -3.22                                                                                                                    | -1.58                                              | -1.53                                              | -1.55                                              |
| Fronnau                                                                                                                                                                                                                          | Jan                                                                | ity                                                                                                                                                            | 100                                                                            | -1.45                                                                                                           | 102                                                                                  | -1.00                                                                                                            | 179                                                                                        | -1.00                                                                                                                    |                                                    |                                                    |                                                    |
| Gross                                                                                                                                                                                                                            | Jan                                                                | base                                                                                                                                                           | 204                                                                            | -2 55                                                                                                           | 204                                                                                  | -2 55                                                                                                            | 204                                                                                        | -2 55                                                                                                                    | -2.38                                              | -2.38                                              | -2.36                                              |
| Glienicke                                                                                                                                                                                                                        | Juli                                                               | 5450                                                                                                                                                           | 201                                                                            | 2.00                                                                                                            | 201                                                                                  | 2.00                                                                                                             | 201                                                                                        | 2.00                                                                                                                     | 2.00                                               | 2.00                                               | 2.00                                               |
| Gross                                                                                                                                                                                                                            | Jan                                                                | sensitiv                                                                                                                                                       | 228                                                                            | -0.17                                                                                                           | 228                                                                                  | -0.17                                                                                                            | 227                                                                                        | -0.19                                                                                                                    |                                                    |                                                    |                                                    |
| Glienicke                                                                                                                                                                                                                        |                                                                    | ity                                                                                                                                                            |                                                                                |                                                                                                                 |                                                                                      |                                                                                                                  |                                                                                            |                                                                                                                          |                                                    |                                                    |                                                    |
| Grunewald                                                                                                                                                                                                                        | Jan                                                                | base                                                                                                                                                           | 188                                                                            | -5.86                                                                                                           | 188                                                                                  | -5.86                                                                                                            | 188                                                                                        | -5.86                                                                                                                    | -2.88                                              | -2.88                                              | -2.88                                              |
| Grunewald                                                                                                                                                                                                                        | Jan                                                                | sensitiv                                                                                                                                                       | 212                                                                            | -2.98                                                                                                           | 212                                                                                  | -2.98                                                                                                            | 212                                                                                        | -2.98                                                                                                                    |                                                    |                                                    |                                                    |
|                                                                                                                                                                                                                                  |                                                                    | ity                                                                                                                                                            |                                                                                |                                                                                                                 |                                                                                      |                                                                                                                  |                                                                                            |                                                                                                                          |                                                    |                                                    |                                                    |
| J. u. W.                                                                                                                                                                                                                         | Jan                                                                | base                                                                                                                                                           | 69                                                                             | -0.19                                                                                                           | 69                                                                                   | -0.19                                                                                                            | 67                                                                                         | -0.46                                                                                                                    | -1.67                                              | -1.67                                              | -1.72                                              |
| Brauer Platz                                                                                                                                                                                                                     |                                                                    |                                                                                                                                                                |                                                                                |                                                                                                                 |                                                                                      |                                                                                                                  |                                                                                            |                                                                                                                          |                                                    |                                                    |                                                    |
| J. u. W.
Brauer Platz                                                                                                                                                                                                         | Jan                                                                | sensitiv
itv                                                                                                                                                | 69                                                                             | 1.48                                                                                                            | 69                                                                                   | 1.48                                                                                                             | 67                                                                                         | 1.26                                                                                                                     |                                                    |                                                    |                                                    |
| Mueggelsee                                                                                                                                                                                                                       | Jan                                                                | base                                                                                                                                                           | 64                                                                             | 5.59                                                                                                            | 64                                                                                   | 5.59                                                                                                             | 64                                                                                         | 5.59                                                                                                                     | 0.00                                               | 0.00                                               | 0.00                                               |
| damm                                                                                                                                                                                                                             |                                                                    |                                                                                                                                                                |                                                                                |                                                                                                                 |                                                                                      |                                                                                                                  |                                                                                            |                                                                                                                          |                                                    |                                                    |                                                    |
| Mueggelsee                                                                                                                                                                                                                       | Jan                                                                | sensitiv                                                                                                                                                       | 64                                                                             | 5.59                                                                                                            | 64                                                                                   | 5.59                                                                                                             | 64                                                                                         | 5.59                                                                                                                     |                                                    |                                                    |                                                    |
| damm                                                                                                                                                                                                                             |                                                                    | ity                                                                                                                                                            |                                                                                |                                                                                                                 |                                                                                      |                                                                                                                  |                                                                                            |                                                                                                                          |                                                    | 0.17                                               |                                                    |
| Schichauwe                                                                                                                                                                                                                       | Jan                                                                | base                                                                                                                                                           | 34                                                                             | 2.78                                                                                                            | 33                                                                                   | 2.76                                                                                                             | 31                                                                                         | 2.47                                                                                                                     | -2.36                                              | -2.47                                              | -2.18                                              |
| g                                                                                                                                                                                                                         | lan                                                                |                                                                                                                                                                | 24                                                                             | E 1 4                                                                                                           | 22                                                                                   | 5.00                                                                                                             | 20                                                                                         | 4.04                                                                                                                     |                                                    |                                                    |                                                    |
| Schichauwe                                                                                                                                                                                                                       | Jan                                                                | sensitiv                                                                                                                                                       | 34                                                                             | 5.14                                                                                                            | 33                                                                                   | 5.22                                                                                                             | 30                                                                                         | 4.04                                                                                                                     |                                                    |                                                    |                                                    |
|                                                                                                                                                                                                                                  |                                                                    | ity /                                                                                                                                                          |                                                                                |                                                                                                                 |                                                                                      |                                                                                                                  |                                                                                            |                                                                                                                          |                                                    |                                                    |                                                    |
| 9
Blankenfeld                                                                                                                                                                                                                 | hul                                                                | ity
base                                                                                                                                                    | 157                                                                            | 0.25                                                                                                            | 151                                                                                  | -0.01                                                                                                            | 144                                                                                        | 0.15                                                                                                                     | -0.24                                              | -0.42                                              | -0.32                                              |
| 9
Blankenfeld
e-Mahlow                                                                                                                                                                                                     | Jul                                                                | ity
base                                                                                                                                                    | 157                                                                            | 0.25                                                                                                            | 151                                                                                  | -0.01                                                                                                            | 144                                                                                        | 0.15                                                                                                                     | -0.24                                              | -0.42                                              | -0.32                                              |
| 9
Blankenfeld
e-Mahlow
Blankenfeld                                                                                                                                                                                      | Jul
Jul                                                         | ity
base
sensitiv                                                                                                                                        | 157
173                                                                     | 0.25                                                                                                            | 151
170                                                                           | -0.01
0.41                                                                                                    | 144
163                                                                                 | 0.15                                                                                                                     | -0.24                                              | -0.42                                              | -0.32                                              |
| Blankenfeld
e-Mahlow
Blankenfeld
e-Mahlow                                                                                                                                                                               | Jul
Jul                                                         | ity
base
sensitiv
ity                                                                                                                                 | 157
173                                                                     | 0.25
0.48                                                                                                    | 151
170                                                                           | -0.01
0.41                                                                                                    | 144
163                                                                                 | 0.15
0.48                                                                                                             | -0.24                                              | -0.42                                              | -0.32                                              |
| g
Blankenfeld
e-Mahlow
Blankenfeld
e-Mahlow
Buch                                                                                                                                                                  | Jul
Jul
Jul                                                  | ity
base
sensitiv
ity
base                                                                                                                         | 157
173
47                                                               | 0.25
0.48
-2.69                                                                                           | 151
170
46                                                                     | -0.01
0.41
-2.56                                                                                           | 144
163
44                                                                           | 0.15
0.48
-2.38                                                                                                    | -0.24                                              | -0.42                                              | -0.32
-0.89                                     |
| g
Blankenfeld
e-Mahlow
Blankenfeld
e-Mahlow
Buch
Buch                                                                                                                                                          | Jul
Jul
Jul
Jul                                           | ity
base
sensitiv
ity
base
sensitiv                                                                                                             | 157
173
47
50                                                         | 0.25
0.48
-2.69
-1.58                                                                                  | 151
170
46
49                                                               | -0.01
0.41
-2.56
-1.44                                                                                  | 144
163
44
47                                                                     | 0.15
0.48
-2.38
-1.49                                                                                           | -0.24                                              | -0.42                                              | -0.32
-0.89                                     |
| g
Blankenfeld
e-Mahlow
Blankenfeld
e-Mahlow
Buch
Buch                                                                                                                                                          | Jul
Jul
Jul
Jul                                           | ity
base
sensitiv
ity
base
sensitiv
ity                                                                                                      | 157
173
47
50                                                         | 0.25
0.48
-2.69
-1.58                                                                                  | 151
170
46
49                                                               | -0.01
0.41
-2.56
-1.44                                                                                  | 144
163
44
47                                                                     | 0.15
0.48
-2.38
-1.49                                                                                           | -0.24                                              | -0.42                                              | -0.32                                              |
| g
Blankenfeld
e-Mahlow
Blankenfeld
e-Mahlow
Buch
Buch
Frohnau                                                                                                                                               | Jul
Jul
Jul
Jul
Jul                                    | ity
base
sensitiv
ity
base
sensitiv
ity
base                                                                                              | 157
173
47
50
41                                                   | 0.25
0.48
-2.69
-1.58
-2.25                                                                         | 151
170
46
49
41                                                         | -0.01
0.41
-2.56
-1.44
-2.25                                                                         | 144
163
44
47
40                                                               | 0.15
0.48
-2.38
-1.49
-2.26                                                                                  | -0.24
-1.11
-1.96                            | -0.42
-1.12
-1.56                            | -0.32
-0.89
-0.65                            |
| g
Blankenfeld
e-Mahlow
Blankenfeld
e-Mahlow
Buch
Buch
Frohnau
Frohnau                                                                                                                                    | Jul
Jul
Jul
Jul
Jul
Jul                             | ity
base
sensitiv
ity
base
sensitiv
ity
base
sensitiv                                                                                  | 157
173
47
50
41
44                                             | 0.25
0.48
-2.69
-1.58
-2.25
-0.28                                                                | 151
170
46
49
41
43                                                   | -0.01
0.41
-2.56
-1.44
-2.25
-0.69                                                                | 144
163
44
47
40
41                                                         | 0.15
0.48
-2.38
-1.49
-2.26
-1.60                                                                         | -0.24
-1.11
-1.96                            | -0.42
-1.12
-1.56                            | -0.32
-0.89
-0.65                            |
| y
Blankenfeld
e-Mahlow
Blankenfeld
e-Mahlow
Buch
Buch
Frohnau
Frohnau                                                                                                                                    | Jul
Jul
Jul
Jul
Jul
Jul                             | ity
base
sensitiv
ity
base
sensitiv
ity
base
sensitiv
ity                                                                           | 157
173
47
50
41
44                                             | 0.25
0.48
-2.69
-1.58
-2.25
-0.28                                                                | 151
170
46
49
41
43                                                   | -0.01
0.41
-2.56
-1.44
-2.25
-0.69                                                                | 144
163
44
47
40
41                                                         | 0.15
0.48
-2.38
-1.49
-2.26
-1.60                                                                         | -0.24
-1.11
-1.96                            | -0.42
-1.12
-1.56                            | -0.32
-0.89
-0.65                            |
| g
Blankenfeld
e-Mahlow
Blankenfeld
e-Mahlow
Buch
Buch
Frohnau
Frohnau
Gross                                                                                                                           | Jul
Jul
Jul
Jul
Jul
Jul
Jul                      | ity
base
sensitiv
ity
base
sensitiv
ity
base
sensitiv
ity
base                                                                   | 157

97                                       | 0.25
0.48
-2.69
-1.58
-2.25
-0.28
-4.43                                                       | 151

91                                             | -0.01
0.41
-2.56
-1.44
-2.25
-0.69
-5.64                                                       | 144

87                                                   | 0.15
0.48
-2.38
-1.49
-2.26
-1.60
-6.35                                                                | -0.24
-1.11
-1.96
-0.67                   | -0.42
-1.12
-1.56
-1.35                   | -0.32
-0.89
-0.65
-1.24                   |
| y
Blankenfeld
e-Mahlow
Blankenfeld
e-Mahlow
Buch
Buch
Frohnau
Frohnau
Gross
Glienicke                                                                                                              | Jul
Jul
Jul
Jul
Jul
Jul
Jul                      | ity
base
sensitiv
ity
base
sensitiv
ity
base
sensitiv
ity
base                                                                   | 157

97                                       | 0.25
0.48
-2.69
-1.58
-2.25
-0.28
-4.43                                                       | 151

91                                             | -0.01
0.41
-2.56
-1.44
-2.25
-0.69
-5.64                                                       | 144

87                                             | 0.15
0.48
-2.38
-1.49
-2.26
-1.60
-6.35                                                                | -0.24
-1.11
-1.96
-0.67                   | -0.42
-1.12
-1.56
-1.35                   | -0.32
-0.89
-0.65
-1.24                   |
| y
Blankenfeld
e-Mahlow
Blankenfeld
e-Mahlow
Buch
Buch
Frohnau
Frohnau
Gross
Glienicke
Gross                                                                                                     | Jul
Jul
Jul
Jul
Jul
Jul
Jul
Jul               | ity
base
sensitiv
ity
base
sensitiv
ity
base
sensitiv
ity
base                                                                   | 157

96                                 | 0.25
0.48
-2.69
-1.58
-2.25
-0.28
-0.28
-4.43
-3.76                                     | 151

94                                       | -0.01
0.41
-2.56
-1.44
-2.25
-0.69
-5.64
-4.30                                              | 144

89                                             | 0.15
0.48
-2.38
-1.49
-2.26
-1.60
-6.35
-5.12                                                       | -0.24
-1.11
-1.96
-0.67                   | -0.42
-1.12
-1.56
-1.35                   | -0.32
-0.89
-0.65
-1.24                   |
| y
Blankenfeld
e-Mahlow
Blankenfeld
e-Mahlow
Buch
Buch
Frohnau
Frohnau
Gross
Glienicke
Gross
Glienicke                                                                                        | Jul
Jul
Jul
Jul
Jul
Jul
Jul
Jul               | ity
base
sensitiv
ity
base
sensitiv
ity
base
sensitiv
ity
base                                                                   | 157

97                           | 0.25
0.48
-2.69
-1.58
-2.25
-0.28
-4.43
-3.76                                              | 151

94                                 | -0.01
0.41
-2.56
-1.44
-2.25
-0.69
-5.64
-4.30                                              | 144

89                                       | 0.15
0.48
-2.38
-1.49
-2.26
-1.60
-6.35
-5.12
-4.63                                              | -0.24
-1.11
-1.96
-0.67                   | -0.42
-1.12
-1.56
-1.35
-1.63          | -0.32
-0.89
-0.65
-1.24                   |
| y
Blankenfeld
e-Mahlow
Blankenfeld
e-Mahlow
Buch
Buch
Frohnau
Frohnau
Gross
Glienicke
Gross
Glienicke
Grunewald                                                                           | Jul
Jul
Jul
Jul
Jul
Jul
Jul
Jul
Jul        | ity
base
sensitiv
ity
base
sensitiv
ity
base
sensitiv
ity
base
sensitiv
ity
base                                        | 157

95               | 0.25
0.48
-2.69
-1.58
-2.25
-0.28
-0.28
-4.43
-3.76
-3.70
-1.59                   | 151

92                     | -0.01
0.41
-2.56
-1.44
-2.25
-0.69
-5.64
-4.30
-3.86
-2.23                            | 144

89                     | 0.15
0.48
-2.38
-1.49
-2.26
-1.60
-6.35
-5.12
-4.63
-2.03                                     | -0.24
-1.11
-1.96
-0.67
-2.11          | -0.42
-1.12
-1.56
-1.35
-1.63          | -0.32
-0.89
-0.65
-1.24
-2.59          |
| g
Blankenfeld
e-Mahlow
Blankenfeld
e-Mahlow
Buch
Buch
Buch
Frohnau
Frohnau
Gross
Glienicke
Gross
Glienicke
Grunewald
Grunewald                                                      | Jul
Jul
Jul
Jul
Jul
Jul
Jul
Jul
Jul
Jul | ity
base
sensitiv
ity
base
sensitiv
ity
base
sensitiv
ity
base
sensitiv
ity
base                                        | 157

95                     | 0.25
0.48
-2.69
-1.58
-2.25
-0.28
-4.43
-3.76
-3.70
-1.59                            | 151

92                     | -0.01
0.41
-2.56
-1.44
-2.25
-0.69
-5.64
-4.30
-3.86
-2.23                            | 144

89                     | 0.15
0.48
-2.38
-1.49
-2.26
-1.60
-6.35
-5.12
-5.12
-4.63
-2.03                            | -0.24
-1.11
-1.96
-0.67
-2.11          | -0.42
-1.12
-1.56
-1.35
-1.63          | -0.32
-0.89
-0.65
-1.24
-2.59          |
| g
Blankenfeld
e-Mahlow
Blankenfeld
e-Mahlow
Buch
Buch
Buch
Frohnau
Frohnau
Gross
Glienicke
Gross
Glienicke
Grunewald
Grunewald                                                      | Jul
Jul
Jul
Jul
Jul
Jul
Jul
Jul
Jul
Jul | ity
base
sensitiv
ity
base
sensitiv
ity
base
sensitiv
ity
base
sensitiv
ity
base                                        | 157

158        | 0.25
0.48
-2.69
-1.58
-2.25
-0.28
-0.28
-0.28
-3.70
-3.70
-1.59
-3.37          | 151

149              | -0.01
0.41
-2.56
-1.44
-2.25
-0.69
-5.64
-4.30
-3.86
-2.23
-3.95                   | 144

142              | 0.15
0.48
-2.38
-1.49
-2.26
-1.60
-6.35
-5.12
-4.63
-2.03
-4.50                            | -0.24
-1.11
-1.96
-0.67
-2.11
-1.28 | -0.42
-1.12
-1.56
-1.35
-1.63
-1.55 | -0.32
-0.89
-0.65
-1.24
-2.59
-1.02 |
| y
Blankenfeld
e-Mahlow
Blankenfeld
e-Mahlow
Buch
Buch
Buch
Frohnau
Frohnau
Gross
Glienicke
Gross
Glienicke
Grunewald
Grunewald
J. u. W.
Brauer Platz                          | Jul
Jul
Jul
Jul
Jul
Jul
Jul
Jul
Jul
Jul | ity
base
sensitiv
ity
base
sensitiv
ity
base
sensitiv
ity
base
sensitiv
ity
base                                        | 157

158        | 0.25
0.48
-2.69
-1.58
-2.25
-0.28
-0.28
-4.43
-3.76
-3.70
-1.59
-3.37          | 151

149              | -0.01
0.41
-2.56
-1.44
-2.25
-0.69
-5.64
-4.30
-3.86
-2.23
-3.95                   | 144

142              | 0.15
0.48
-2.38
-1.49
-2.26
-1.60
-6.35
-5.12
-4.63
-2.03
-4.50                            | -0.24
-1.11
-1.96
-0.67
-2.11
-1.28 | -0.42
-1.12
-1.56
-1.35
-1.63
-1.55 | -0.32
-0.89
-0.65
-1.24
-2.59
-1.02 |
| g
Blankenfeld
e-Mahlow
Blankenfeld
e-Mahlow
Buch
Buch
Frohnau
Frohnau
Gross
Glienicke
Gross
Glienicke
Grunewald
Grunewald
Grunewald
J. u. W.
Brauer Platz
J. u. W.         | Jul
Jul
Jul
Jul
Jul
Jul
Jul
Jul
Jul
Jul | ity
base
sensitiv
ity
base
sensitiv
ity
base
sensitiv
ity
base
sensitiv
ity
base
sensitiv
ity
base
sensitiv | 157

158 | 0.25
0.48
-2.69
-1.58
-2.25
-0.28
-0.28
-0.28
-3.70
-3.70
-1.59
-3.37
-3.37 | 151

152 | -0.01
0.41
-2.56
-1.44
-2.25
-0.69
-3.64
-4.30
-3.86
-2.23
-3.95
-2.40          | 144

144 | 0.15
0.48
-2.38
-1.49
-2.26
-1.60
-1.60
-6.35
-5.12
-4.63
-2.03
-4.50
-4.50
-3.48 | -0.24
-1.11
-1.96
-0.67
-2.11
-1.28 | -0.42
-1.12
-1.56
-1.35
-1.63
-1.55 | -0.32
-0.89
-0.65
-1.24
-2.59
-1.02 |
| g
Blankenfeld
e-Mahlow
Blankenfeld
e-Mahlow
Buch
Buch
Buch
Frohnau
Frohnau
Gross
Glienicke
Gross
Glienicke
Grunewald
Grunewald
Grunewald
J. u. W.
Brauer Platz
J. u. W. | Jul
Jul
Jul
Jul
Jul
Jul
Jul
Jul
Jul
Jul | ity
base
sensitiv
ity
base
sensitiv
ity
base
sensitiv
ity
base
sensitiv
ity
base
sensitiv
ity
base             | 157

158 | 0.25
0.48
-2.69
-1.58
-2.25
-0.28
-0.28
-3.70
-1.59
-3.70
-1.59
-3.37
-2.09 | 151

152       | -0.01
0.41
-2.56
-1.44
-2.25
-0.69
-5.64
-4.30
-3.86
-2.23
-3.95
-3.95
-2.40 | 144

144 | 0.15
0.48
-2.38
-1.49
-2.26
-1.60
-6.35
-5.12
-4.63
-2.03
-4.50
-4.50
-3.48          | -0.24
-1.11
-1.96
-0.67
-2.11
-1.28 | -0.42
-1.12
-1.56
-1.35
-1.63
-1.55 | -0.32
-0.89
-0.65
-1.24
-2.59
-1.02 |

| damm       |     |          |     |       |     |       |     |       |       |       |       |
|------------|-----|----------|-----|-------|-----|-------|-----|-------|-------|-------|-------|
| Mueggelsee | Jul | sensitiv | 160 | 0.09  | 154 | -0.29 | 148 | -1.08 |       |       |       |
| damm       |     | ity      |     |       |     |       |     |       |       |       |       |
| Schichauwe | Jul | base     | 143 | -3.54 | 137 | -3.64 | 130 | -3.88 | -0.65 | -0.85 | -0.80 |
| g          |     |          |     |       |     |       |     |       |       |       |       |
| Schichauwe | Jul | sensitiv | 169 | -2.88 | 158 | -2.79 | 151 | -3.08 |       |       |       |
| g          |     | ity      |     |       |     |       |     |       |       |       |       |

However, a small imprecision in the reporting of the results needs to be corrected in the manuscript, see below.

**Changes in the manuscript**

P. 17, line 20: "With some differences between the stations, the bias of weekday downwind NO2 concentrations was reduced by between ca. 1.5 and  $3.02.9 \ \mu g$  m-3 January and ca. 0.74 and 1.5  $\mu g$  m-3 in July."

**Referee 2**

In its current form, it shows a case study for the city of Berlin which maybe should be included in the title. It would be nice if the authors could add a section or a short passage, discussing the questions What do we learn from this study? Are the results transferrable to other urban areas?

P21: Some open questions should be briefly discussed her as mentioned in the beginning: Why is the study important for the field of research? What were the novel findings? What can we learn? Are the results transferrable to other urban areas?

**Answer**

In agreement with the reviewer, we add to the conclusions as follows:

**Changes in the manuscript**

Page 21, line 15: "The still negative bias is consistent with the factor being a rather conservative estimate.

The emission inventory used in this study is based on officially reported emissions by the individual countries, and the emissions are spatially distributed by TNO based on proxy data. Assuming the quality and accuracy of the proxy data is similar, at least for larger German cities, and considering that modelling studies for other German cities have also shown an underestimation of simulated NO2 concentrations using the same emission inventory, we would assume that the results found in this study for Berlin may generally be transferrable to at least other German metropolitan areas. The underestimation of NO2 concentrations throughout the day, the consistency of the calculated correction factor with findings from other studies and the improvement of model results applying the correction factor suggest that more research is needed in order to more accurately understand the spatial and temporal variability in real-world NOx emissions from traffic, and apply this understanding to the inventories used in high resolution chemical transport models. Given the above considerations, this not only holds for the urban area of Berlin, but for German and most

likely European metropolitan areas more generally."

**Specific Comments**

**Referee 1**

It might be helpful to expand the abbreviations MAM, JJA, SON, and DJF when they first appear, to make the paper more accessible to those who are not familiar with these common abbreviations.

**Answer**

Manuscript updated as suggested.

**Changes in the manuscript**

The captions of Tables 3 and 4 as updated to include the explanation of abbreviations: "Data are aggregated as follows: MAM - March, April, May, JJA - June, July, August, SON - September, October, November, DJF - December, January, February."

**Referee 2**

Page 1: Line 11: It is unclear at this point you have treated the discrepancy between model resolution and resolution of the emission data. What is the purpose of the emission downscaling to 1km and what was the initial resolution of the inventory. These points should be discussed more details in the methodology section, also explaining the downscaling method.

P4: 16: see comment above about the purpose of downscaling

P6: 2: specify chosen categories and reason for the selection

**Answer**

Since the original resolution of the emission inventory is much coarser (7km x 7km) than the model resolution used in this study (3km x 3km) we find it necessary to downscale the emission categories most relevant for this study based on representative proxy data. Kuik et al. (2016) have shown that this helps to better capture the spatial distribution of pollutant concentrations. Since the downscaling of the emissions was developed for the study presented in Kuik et al. (2016), which also included model simulations at a horizontal resolution of 1kmx1km, the emissions were downscaled to 1kmx1km instead of 3kmx3km.The inventory, reason and process of downscaling are discussed in Section 2.4 (extended as specified below) and in Kuik et al., 2016.

**Changes in the manuscript**

Page 1, line 11: "The emission data are downscaled from an original resolution of ca. 7 km x 7 km to a resolution of 1 km x 1 km"

Page 6, line 4, "In addition, we only downscale those emission categories (SNAP categories) which are both of main interest for studying NO2 in an urban area and also represented well by the proxy data chosen. This ensures that we are not suggesting a higher precision than achievable with the available proxy data. We thus only downscale emissions from SNAP

categories 2 (residential combustion), 6 (product use) and 71-75 (traffic), as these emissions can be represented well by population density (SNAP 2 and 6) and traffic density (SNAP 71-75)."

**Referee 2**

P2: 9ff: better EEA 21: better WHO

Referee 3 Page 13, line 9: Eea and Wmo should be capitalized

Answer

Manuscript updated as suggested.

**Changes in the manuscript**

EEA and WHO capitalized throughout the manuscript.

**Referee 2**

Page 2: 10: What is the reason for the downward trend?

**Answer**

The reference given in the manuscript (EEA 2016) attributes the downward trend to decreasing NOx emissions. We add this information to the manuscript.

**Changes in the manuscript**

Page, line 10: "While there is a downward trend in NO2 concentrations due to decreasing NOx emissions, [...]"

**Referee 2**

Page 2: 16: specify 'real world driving condition'

**Answer**

We meant conditions that resemble typical driving situations either because they are simulated accurately in the lab or because the car is actually driving on the road. To clarify, we rephrase as follows:

**Changes in the manuscript**

"[...] despite increasingly strict emission standards for diesel cars with the introduction of the Euro 5 and Euro 6 norms, in real-world driving conditions-under real-world driving conditions, i.e. the pollutants a car produces while being driven on real roads as opposed to being tested in a lab, Euro 5-certified cars exceed the emission limit of [...]"

**Referee 1**

page 3, line 7: it would be more correct to use "and" instead of "with" in this sentence

**Answer**

Manuscript updated as suggested.

**Changes in the manuscript**

Page 3, line 7: "However, many modelling studies report discrepancies between modelled and observed  $NO_2$  concentrations [...]"

**Referee 2**

P3: 3,7: Fallmann et al. 2016 presents a modelling study for the city of Stuttgart following similar procedures and gets similar findings

**Answer**

Thank you for the suggestion; we included this reference in the manuscript.

**Changes in the manuscript**

Page 3, line 20, "Fallmann et al. (2016) report a negative bias in NO2 concentrations simulated with WRF-Chem at 3kmx3km of ca. 50% on average and up to 60% during daytime."

**Referee 2**

P3, 21: unclear: 'basing'

**Answer**

We rephrase:

**Changes in the manuscript**

"Degraeuwe et al. (2016) assess the impact of different diesel NOx emission scenarios on air quality in Antwerp, basing street canyon modelling on urban background concentrations modelled with combining model simulations with LOTOS-EUROS at a horizontal resolution of ca. 7 km x 7 km (urban background) with a street canyon model."

**Referee 2**

P4: 1: what is meant by 'activity data'

**Answer**

We mean data describing the intensity of the (anthropogenic) activity causing the emissions, e.g. fuel burnt. We include this example to illustrate it:

**Changes in the manuscript**

Page 4, line 1:"Emissions are typically estimated from a combination of activity data (e.g. fuel burnt) and emission factors."

**Referee 2**

P4: 5: where does the large error range come from and how did you consider this in your study?

P15: 16: what are the dominant non-traffic emission sources in the area of Berlin in summer time?

**Answer**

As we explain in the manuscript the TNO-MACC III emission inventory used here is based on officially reported emissions and thus consists of the reporting countries' best guess of emissions. The aim of this paper is to use these officially reported numbers and assess how much they might underestimate NO2 emissions in the case of Berlin. The emission error range is thus considered as a range in which the actual emissions are expected to fall.

In summer, traffic emissions in Berlin constitute ca. 60% of all NOx emissions according to the TNO-MACC III inventory. The next largest emission source is emissions from (energy) industry. However, it is important to note that, unlike traffic emissions, energy industry emissions are not all directly at the surface, but distributed between the third and seventh model layer (that is, above ca. 95m).

Both points are made clearer in the revised manuscript by adding the following:

**Changes in the manuscript**

Page 4, line 5, "Emission factors for road transport, for example, may have an error range between 50% and 200%, while emission factors for energy industry emissions, the second largest source of NOx emissions in Berlin, is much better constrained with an error range between 20% and 60% (Kuenen et al., 2014). Emission error ranges for the TNO-MACC III inventory used in this study are determined following the EEA Emission Inventory Guidebook, and depend, for example, on the number of measurements made for deriving the emission factor (EEA, 2013, Kuenen et al., 2014)."

Referee 2 P4:14: better 'model setup'

Answer Manuscript changed as suggested.

Changes in the manuscript

"The model setup, [...]"

**Referee 2**

P4: 21: State, whether the sensitivity simulations are taken from the output of the 1 year run or whether they are independent experiments.

**Answer**

To clarify, we change the manuscript as follows:

**Changes in the manuscript**

Page 4, line 21: "The factor is then tested in two individual one month long sensitivity simulations for January and July 2014, in the following referred to as sensitivity simulations."

**Referee 2**

P4: General: specify the source and the characteristics of the urban canopy parameters

**Answer**

Thank you for your comment. The source and characteristics of the parameters are described in detail in Kuik et al. (2016). We would like to refer the reader to this description (e.g. as indicated on page 4, line 30, 31 in the manuscript), in order to avoid making this present manuscript too long.

**Changes in the manuscript**

None

**Referee 2**

Page 14, line 5 onwards: What is the model height for the NOx evaluation? How do model and observation height compare?

**Referee 3**

Page 4, line 29: 'top' should be added after 'model layer'. With a lowest layer depth of 30 m the near surface profiles are not well resolved. How does this affect the simulated near-surface  $NO_X$  concentrations?

**Answer**

Modelled NOx concentrations are grid cell averages, with the first model layer top at 30m. Measurements are done at a height of 3m.

We agree with Reviewer 3 that the surface profile in the lower 30m is not well resolved. We have tested extrapolating the simulated profiles of NOx concentrations to the surface. Extrapolated daytime concentrations typically differed very little from the grid cell averages, while extrapolated nighttime concentrations would generally be much higher than the simulated grid-cell average. This is due to the steep gradient of the simulated NOx profile during nighttime when the planetary boundary layer is shallow. Ideally, the simulated profiles would be compared to measurements, which could then also be used to extrapolate the modelled grid-cell averages to the ground. Unfortunately such measurements are not available.

We discuss challenges in the comparability of grid cell averages with point measurements on page 13, line 7, and the evaluation of modelled vertical profiles with measurements on page 14, line 19/20 and page 20, line 29/30. We changed the manuscript as suggested by Reviewer 3, and add a sentence emphasizing the use of measured vertical profiles when

modelled profiles are not resolved in the lowest layer.

**Changes in the manuscript**

Page 4, line 29, "[...] with the first model layer top at [...]".

Page 14, line 18: "Overall, this discussion shows that the representation of vertical mixing over urban areas might have to be improved to be physically more consistent in regional models, for example by better taking into account urban heat and momentum fluxes and treating the urban parameterization consistently with chemistry. Measurements of vertical profiles of NOx in cities, particularly in the planetary boundary layer, would be helpful in order to evaluate the models and improve the representation of surface NOx concentrations, as the NOx profile in the lowest model layer is not resolved at the model resolution used in this study."

**Referee 2**

P5: 20ff: how is second part of the simulation initialized, how does it refer to the first model period?

**Referee 3**

Page 5, line 21: Why is a 4-day spin-up required for the simulation of the last 6 month of the year?

**Answer**

The simulations for the first and second halves of the year were performed in parallel for reasons of computational efficiency. A comparison of the last days of the first simulation with the (overlapping) first days of the second simulation showed no discontinuities.

**Changes in the manuscript**

Page 5, line 20: "Both simulations are initialized using data from ERA-Interim (meteorology) and MOZART4/GEOS5 (chemistry) and preceded by a spin-up period of 4 days."

**Referee 2**

P5, 28: Discuss briefly if you can estimate an error from using 2011 emission data for a 2014 simulation?

**Answer**

From comparing the TNO-MACC III emissions for Germany in the years available, there was generally only a very small (decreasing) trend in reported emissions up to 2011, expected to continue also after 2011. This allows using the latest available year of emissions (2011) also for 2014 simulations (Hugo Denier van der Gon, personal communication). Thus, we expect the error due to the different year of emissions to be much smaller than the uncertainty related to the quantification of traffic NOx emissions. The latter is also supported by other studies reporting similar results using different methods (and different times) to assess the difference between reported and actual NOx emissions.

**Changes in the manuscript**

Page 5, line 26: "[...] The latest available year is 2011, which we use for simulating the year

2014. From comparing the TNO-MACC III emissions for Germany in the years available, there was generally only a very small (decreasing) trend in reported emissions up to 2011, expected to continue also after 2011. This allows using the latest available year of emissions (2011) also for 2014 simulations (Hugo Denier van der Gon, personal communication)."

**Referee 3**

Page 6, line 9: Please mention also the heights and not also the layer numbers.

**Answer**

We include the approximate layer heights in addition. Due to the terrain-following pressure coordinates of WRF, the exact value of the heights varies slightly.

**Changes in the manuscript**

P. 6, line 11: "[...] are distributed vertically into the first seven layers (see Supplementary Material for further details). For reference, the layer tops are at ca. 30 m (layer 1), 95 m (layer 2), 190 m (layer 3), 310 m (layer 4), 460 m (layer 5), 650 m (layer 6) and 890 m (layer 7)."

**Referee 3**

Page 6, line 28: Does the percentage refer to NO2 mass? Please clarify.

**Answer**

Yes, it does. We include this in the manuscript as follows:

**Changes in the manuscript**

Page 6, line 28, "[...] NOx is emitted as 10% NO2 and 90% NO (by mass)."

**Referee 2**

P6, 23: Can you give an estimate of the error in mean values when only considering weekdays?

P13 30/31: earlier it was mentioned that only weekday profiles were applied in the model setup. Please clarify this aspect here.

**Referee 3**

Page 6, line 23: Why is the same diurnal cycle applied for weekdays and weekends? The traffic counts show certainly a different course for weekdays and weekends.

**Answer**

Weekly profiles were applied (depending on the day of the week), and daily profiles were applied (depending on the hour of the day). Applied diurnal profiles are not weekday-dependent, but calculated from averaging the profiles of all days of the week (and stations). We clarify this in the manuscript.

The reason for applying the same diurnal cycle on weekends and weekdays is mainly

computational efficiency, as the emission processing applied here is already very complex. Furthermore, this processing is in line with the way emissions are processed in many other modelling studies. As the applied average diurnal cycle is very similar to the weekday diurnal cycle, we expect that this would mainly effect simulated weekend concentrations. Furthermore, a comparison of the root mean square error of the (station-and-weekday-) average diurnal cycle with the (station-) average weekday and weekend diurnal cycles shows that the error is comparably small during daytime, which is the time period of main interest for this study. However, we agree that a distinction between weekend and weekday diurnal cycles might potentially be an improvement of the model setup. This would be part of the improvements in traffic emission diurnal cycles discussed in the conclusions of the manuscript.

**Changes in the manuscript**

P. 6, line 22: "[...] we apply a uniform diurnal cycle for each day of the week, making no distinction between the diurnal cycle of weekends and weekdays. As mentioned above, we do however apply also a weekly profile, thus the magnitude of the daily emissions on weekends is different from that on weekdays."

**Referee 2**

P7: 25: Procedure of NO2 retrieval at Air Base station unclear. How do you get the final NO2 concentrations from converted NO?

**Answer**

We clarify this in the manuscript as follows:

**Changes in the manuscript**

Page 7, line 11: "[...] as required by EU clean air legislation. The files can directly be downloaded from the AirBase website."

Page 7, line 25: "NO2 concentrations used for this study were measured using chemiluminescence. With this method, NO2 is converted to NO with a molybdenum converter before being detected using chemiluminescence, as NO reacts with O3 to form NO2 and O2 while emitting light (see, e.g. Gerboles et al., 2003, and Steinbacher et al., 2007)."

**Referee 2**

P8: 20,21: Specify more details about 'using the model setup for [. . .] policy relevant'. Do you mean operational forecast here?

**Answer**

Rather than operational forecasts, modelling studies aiming at identifying and/or evaluating NO2 reduction measures are meant.

**Changes in the manuscript**

Page 8, line 20/21: "Modelled NO2 concentrations are evaluated with the aim of using the model setup for policy-relevant analyses of urban NO2 concentrations and NO2 reduction measures with high temporal and spatial resolution, [...]"

**Referee 3**

Page 8, line 25: The equation for MQO should be inserted already here.

**Answer**

Manuscript updated as suggested.

**Changes in the manuscript**

Page 8, line 26: Equation 2 is inserted here.

**Referee 1**

page 8, 2nd paragraph: urRV needs proper formatting twice here

**Answer**

Manuscript updated as suggested.

**Changes in the manuscript**

Page 8, lines 7 and 9: Formatting corrected.

**Referee 3**

Page 10, line 3: The linking between NO, NO2 and O3 is also true for offline models.

**Answer**

We absolutely agree. We rephrase the corresponding sentence to make the wording clearer.

**Changes in the manuscript**

Page 10, line 1: "[...] we include a brief evaluation of selected key meteorological parameters (temperature, wind speed and direction) as well as further chemical species (O3, NOx), the former because as WRF-Chem is an online-coupled model, and the latter because NO2 is tightly linked to NO and O3."

**Referee 3**

Page 10, line 25: Please add some details about the 'misreprentation of the diurnal cycles'.

**Answer**

We assume this comment relates to page 14, line 25. We clarify this sentence by changing the punctuation.

**Changes in the manuscript**

P. 14, line 25: "This is consistent with an overall underestimation of emission sources active in the morning hours on weekdays and potentially also a misrepresentation of the diurnal cycles of emissions in the model-: Ttraffic emissions are distributed in the model throughout the day using [...]"

**Referee 2**

P10: 25: state briefly why you changed the model setup compared to Kuik et al (2016) with regard to re-initialization etc. Can you discuss the impact of the modifications on NO, NO2, NOx, O3?

**Answer**

With regard to re-initialization, we changed the model setup because the length of the simulation presented here is considerably longer. We therefore decided to follow the procedure used to perform the AQMEII simulations (see manuscript for references). Because of the short spin-up times of the meteorology and chemistry in the model, we expect the impact of the reinitialization technique on the analyzed concentrations of NO, NO2, NOx and O3 to be rather small. The main benefit of this technique is an improved comparability of modelled pollutants with observed time series as the simulated meteorology follows the observations more closely. Please also see our answer to comments of referees 1 and 2 (first comment answered above).

**Changes in the manuscript**

None

**Referee 2**

P10, line 30: reference height for wind speed and temperature? Model height 30m?

**Answer**

Clarified in the manuscript as follows:

**Changes in the manuscript**

Page 10, line 30: "Modelled and observed 2m temperature and 10m wind speed are compared at five stations run by the German Weather Service [...]"

**Referee 2**

P11: 5-21: Indicate briefly the reason for different model performance for different seasons/daytime. Does that relate to general problems with tour model setup (boundary layer scheme, chemical mechanism etc.)

5.1/5.2: how do the biases in meteorology relate to biases in chemical species? Is there a seasonal dependence?

**Answer**

It is widely known that the performance of meteorological variables depend on the area of interest, time of the day, season, land use data, the weather patterns, the PBL and land surface scheme used to set a model configuration etc. To ensure best model performance, multiple simulations would be required, but this approach is beyond the main purpose of this study. Furthermore, best model performance would still not imply an equally good performance throughout all seasons.

On the comment concerning Sect. 5.1/5.2 – we address this question in detail in Sect. 5.3. As mentioned in Sect. 5.3, we find that the model has some difficulties to capture the variation of NO2 concentrations on time scales below 2.5 days, which might be related to problems in modelled mixing. One seasonally dependent bias that is mentioned consists in higher than observed peaks in the model simulation at rural background stations in autumn, which might be related to a bias in wind direction.

**Changes in the manuscript**

None

**Referee 3**

Page 13, line 9: This is the case for all simulations with a grid width of only 3 km.

**Answer**

Yes. We make this clearer by rephrasing the corresponding sentence as follows:

**Changes in the manuscript**

P. 13, line 9: "[...] influenced by local sources that cannot be captured by WRF-Chem run at a horizontal resolution of 3 km x 3 km."

**Referee 2**

P13: 24: How is the link between population density and emission achieved/applied in the model?

**Answer**

The downscaling is described in Section 2.4 of the manuscript. We make the link clearer in the manuscript as follows:

**Changes in the manuscript**

Page 13, line 23: "The re-distribution of these emissions based on population density, as described in Sect. 2.4, may also have contributed to a better spatial representation in our study."

**Referee 2**

P14, 14: better 'too strong' instead of 'too efficient'

**Answer**

Manuscript changed as suggested.

**Changes in the manuscript**

Page 14, line 14: "This might be explained by mixing over urban areas during daytime that is too strong efficient."

**Referee 2**

P14: 5ff: Does a change of the urban canopy model make a change in boundary layer mixing close to the surface? How was the change in model code achieved?

Some boundary layer schemes have been modified recently in order to take into ac- count the mixing of chemical compounds inside the boundary layer. Can you comment on this? What impact might this have to your study?

P14, 15: Did you find a sensitivity to UCM selection?

**Answer**

In this study we have not tested different UCMs. In the WRF-Chem version used here (v3.8.1), the available urban canopy models are the single-layer UCM, and the more complex multi-layer building effect parametrization (BEP) and building energy model (BEM). The multi-layer parametrizations have not been tested for several reasons:

- They would strongly increase computational costs.

- Other studies have shown that they do not outperform simpler approaches when simulating surface temperature.

- They can only be combined with the MYJ and Boulac PBL schemes. The latter cannot be coupled with chemistry, while the former often leads to strong biases in simulated 2m air temperature.

- None of the urban parametrizations are coupled with chemistry, so the only effect on chemistry would be via improved meteorology.

Please also see Kuik et al., 2016, Section 2.3 for a discussion of these issues and further references, and our response to referees 1 and 2 on the subject of recent boundary layer scheme modifications (first comment).

**Changes in the manuscript**

None

**Referee 2**

P14, 25: Is the emission at urban background station generally too low? Maybe you are missing an advection term here?

**Answer**

Looking at the map of NOx emissions (Fig. 1 in the manuscript), we would assume that the biases are not due to an underestimation only at particular grid cells. The map shows that emissions are generally high in the area of the urban background stations (compare with Fig. 2). Advection of chemical species is solved on the same grid and is consist with the simulated meteorological fields. The relatively good model performance for simulated meteorology suggests that advection in the model is fine.

**Changes in the manuscript**

None

**Referee 1**

page 16, line 2: the correction factor should be "f", not "F"?

**Answer**

Manuscript changed as suggested.

**Changes in the manuscript**

Page 16, line 2: Nomenclature of correction factor harmonized ( $f_{NOx}$ ).

**Referee 2**

P16: 18: Are the 'newer' diesel cars already implemented in the inventory used here?

**Answer**

The comment relates to the following sentence in the manuscript: "The seasonal differences might also be influenced by the temperature dependence of NOx emissions in newer diesel cars (Hausberger and Matzer, 2017), leading to higher NOx emissions at colder temperatures, which are not captured by the model." (Page 16, line 18).

The emission inventory/emission processing for the modelling does not take into account a potential temperature-dependence of emissions, while Hausberger and Matzer (2017), also cited in the manuscript, present results for a potential temperature-dependence of NOx emissions of Euro 4, Euro 5 and Euro 6 diesel cars. The contribution of Euro 6 cars in Germany was 1.5% in 2014, Euro 5 accounted for 27% and Euro 4 for 46% (Knörr et al., 2016). A temperature-dependence of their NOx-emissions could contribute to seasonal differences in model performance compared to observations, as described in the manuscript.

**Changes in the manuscript**

None

**Referee 2**

P17: 6: From your experience: Which part of the model/configuration has to be changed for improving the representation of mixing processes?

**Answer**

We address this question in the manuscript in Sect. 5.4 Diurnal and weekly variation of the model bias: "Overall, this discussion shows that the representation of vertical mixing over urban areas might have to be improved to be physically more consistent in regional models, for example by better taking into account urban heat and momentum fluxes and treating the urban parameterization consistently with chemistry." (P. 14, line 18). This point is repeated in the conclusions.

**Changes in the manuscript**

None

**Referee 2**

P17, line 11: Can the difference between secondary and primary NOx be attributed to wrong NO titration processes? Or other chemical processes?

**Answer**

As O3/NOx chemistry is non-linear, we do not think that it is straight forward to attribute the differences to one chemical process. Rather, other processes e.g. the evolution of the boundary layer height also have a strong impact on pollutant concentrations and the chemical regime. Thus, an attribution of the differences would need to look at a variety of different processes and factors, which is beyond the scope of this study. We include this in the manuscript as follows:

**Changes in the manuscript**

Page 17, line 11: "The differences in NO2 and NOx improvements suggest that the impact of the primary NO2 fraction in emitted NOx on observed and modelled NO2 and NOx concentrations, as well as the influence of chemical processes such as NO titration and other relevant physical and chemical processes, might need to be assessed in greater detail."

**Referee 1**

page 17, line 19: should it be "wind direction bin", instead of "wind speed bin"?

**Answer**

Yes, that is true. Changed accordingly.

**Changes in the manuscript**

Page 17, line 18: "[...] exist in the respective wind speed direction bin [...]"

**Referee 2**

P18: 11: The fact that a higher resolution emission inventory does not improve the model results is an important finding here and has to be explained more detailed. Did you run model experiment with a high-resolution emission inventory as well? What is the resolution here?

**Answer**

It appears that the reviewer has misunderstood something here. We do not discuss the resolution of the inventory in this part of the text. Conversely, in a previous study (Kuik et al., 2016), we find that local pollution patterns can be represented better when downscaling the emission inventory from a horizontal resolution of 7km to 1km and increasing the model resolution.

We are glad to further respond to this comment in case the reviewer would like to further elaborate on the comment.

**Changes in the manuscript**

None

**Referee 2**

P18 15-20: What kind of models do you suggest in terms of higher resolution? Chemical transport models, LES, CFD, dispersion models. . ?

**Answer**

The paragraph in the manuscript that this comment refers to speaks about the scaling of roadside NOx concentrations with traffic counts, and tests whether a linear relationship exists (as often assumed in the calculation of the diurnal distribution of traffic emissions). Here, the term "model" is used for mathematical relationships between NOx and traffic counts, not in the context of atmospheric chemistry models. To make it clearer, we replace the term "model" in this section with the term "relationship". We are glad to give a more detailed answer to this question if the reviewer would like to clarify what is being referred to here.

**Changes in the manuscript**

A linear regression model does not explain the variance of observed NOx concentrations at nighttime, as indicated by the R2 close to 0 in Fig. 9. However, during daytime, traffic counts alone explain up to ca. 40% of observed NOx variance, particularly during the

traffic rush hours. The explained variance is smaller during the afternoon peak. In comparison to a linear model relationship, a quadratic model relationship (NOx / (traffic\_count)2) does not explain more of the observed variance (not shown). An exponential model relationship (NOx/exp(traffic\_count)), however, does explain a considerably larger share of the observed variance during daytime and particularly during the traffic rush hours, as depicted in Fig. 9 (up to ca. 60% depending on the station).

**Referee 2**

P18, 25: meaning: HBEFA

**Answer**

In this case yes.

**Changes in the manuscript**

Page 18, line 25, "[...],and emission factors (e.g. from HBEFA) are higher in congested situations compared to free flowing traffic."

**Referee 2**

P20: 27: Urban processes in WRF-Chem are linked via meteorology only. Do you have a suggestion, how UCMs have to be changed in order to improve AQ simulations?

**Answer**

Following the line to which the comment refers, we have already included a suggestion on how such an improvement might be achieved: "For example, the parameterization of urban processes needs to better account for urban heat and momentum fluxes for a more realistic representation of mixing both at daytime and at nighttime, particularly in summer."

**Changes in the manuscript**

None

**Referee 1**

Page 34, figure 8: your description misses out the colour (blue?) of the sensitivity simulation data

**Answer**

Manuscript changed as suggested.

**Changes in the manuscript**

Caption Figure 8: "Time series of hourly observed (black line) and modelled NO2, comparing the base simulation (red) with the sensitivity simulations (blue) [...]"

**Referee 1**

page 38, table 3: in the table caption, the units of MB and RMSE need correcting from ug/m3 to K and m/s (respectively)

**Answer**

Manuscript changed as suggested.

**Changes in the manuscript**

Caption Table 3: "Mean bias (MB) and root mean square error (RMSE) are indicated in K (temperature) and m s-1 (wind speed), [...]"

**Additional minor changes**

P. 21, line 14: "[...] and also improved modelled downwind concentrations."

**References not listed in the manuscript**

- Hu, X.-M., P. M. Klein, and M. Xue (2013), Evaluation of the updated YSU planetary boundary layer scheme within WRF for wind resource and air quality assessments, J. Geophys. Res. Atmos., 118, 10,490–10,505, doi:10.1002/jgrd.50823.
- Knörr, W., Heidt, C., Gores, S., Bergk, F.: Aktualisierung "Daten- und Rechenmodell: Energieverbrauch und Schadstoffemissionen des motorisierten Verkehrs in Deutschland 1960-2035" (TREMOD) für die Emissionsberichterstattung 2016 (Berichtsperiode 1990-2014), https://www.ifeu.de/wpcontent/uploads/Endbericht\_TREMOD\_2016\_160701.pdf (last access: 11 April 2018)

**Top-down quantification of NOx emissions from traffic in an urban area using a high resolution regional atmospheric chemistry model**

Friderike Kuik1,2, Andreas Kerschbaumer3, Axel Lauer4, Aurelia Lupascu1, Erika von Schneidemesser1, and Tim M. Butler1,2

1Institute for Advanced Sustainability Studies, Potsdam, Germany

[revised manuscript text omitted]

$$\mathbf{u}(\mathbf{x}_{i}) = \mathbf{u}_{\mathbf{r}_{\mathbf{R}\mathbf{V}}} \cdot \sqrt{(1-\alpha)\mathbf{x}_{i}^{2} + \alpha \cdot \mathbf{R}\mathbf{V}^{2}}$$
(1)

Here,  $urRV - u_{rev}$  is an estimate of the relative uncertainty around a reference value RV, and  $\alpha$  is the fraction of uncertainty not proportional to the reference value. We use the coefficients corresponding to the mean uncertainties of the individual parameters, i.e.  $urRV u_{rev} = 0.09$ ,  $\alpha = 0.06$  and the reference value RV=200 µg m-3 (Pernigotti et al., 2013).

**3.2 Meteorological data**

- 5 In order to complement the analysis and to investigate potential influences of the modelled meteorology on modelled NO2 concentrations, we include a comparison of modelled meteorology with observations. This includes observations of 2m temperature, and 10m wind speed and direction, all provided by the German Weather Service and available online (Kaspar et al., 2013). In addition, mixing layer height derived from ceilometer measurements at Nansentraße during the BAERLIN2014 campaign (Geiß et al., 2017) are used for a qualitative comparison with the modelled mixing layer height (see Kuik et al., 2016, for
- 10 a discussion of this type of comparison). The data are generally available between 20 June and 27 August 2014, but include a number of gaps.

**4 Analysis and evaluation metrics**

**4.1 Analysis of model results**

Modelled NO2 concentrations are evaluated with the aim of using the model setup for policy-relevant analyses of urban NO2
 concentrations and NO2 reduction measures with high temporal and spatial resolution, and in order to identify the main sources of the errors in modelled NO2 concentrations. For this, we use both operational and diagnostic evaluation metrics, which are explained in the following.

Operational evaluation metrics applied here are based on Thunis et al. (2012) and Pernigotti et al. (2013). They include an analysis of the mean bias (MB) and normalized mean bias (NMB), the correlation coefficient (R), and the root mean square

error (RMSE, as defined in the Supplementary Material). The model error is compared with the model quality objective (MQO) and performance criteria calculated from NO2 observations and their uncertainty. The MQO is defined as follows:

$$MQO = \frac{1}{2} \frac{RMSE}{RMS_U}$$
(2)

Following Thunis et al. (2012) and Pernigotti et al. (2013), a MQO lower than 0.5 indicates that the model results are on average within the range of the measurement uncertainty, and further efforts to improve model performance are not meaningful. A MQO

25 between 0.5 and 1 indicates that the uncertainties of model and observations overlap, and that the model might still be a better predictor of the true value than the observations. A MQO greater than 1, on the other hand, indicates significant differences between the model and the observations. The MQO is defined as follows:

$$MQO = \frac{1}{2} \frac{RMSE}{RMS_U}$$

With  $RMS_U$  being the root mean square of the measurement uncertainty. The performance criteria for mean bias, normalized mean bias and correlation coefficient as defined in Pernigotti et al. (2013) are listed in the Supplementary Material. As the uncertainty of NO2 measurements is partly concentration-dependent, the MQO and the other performance criteria differ between station classes and seasons.

5 The operational evaluation and model quality objectives are intended to support an assessment of the extent to which a model can be used for policy-relevant analyses, but do not point to the underlying processes that might lead to a disagreement between model results and observations. Furthermore, the calculation of the NO2 measurement uncertainty underlying the calculation of the MQO and performance criteria is also based on a number of uncertain parameters.

We thus complement the analysis with a diagnostic evaluation, comparing the individual spectral components of the modelled and observed time series. This is done following Solazzo and Galmarini (2016) and Solazzo et al. (2017): we use a Kolmogorov-Zurbenko filter (Zurbenko, 1986), a widely used filter in the analysis of air quality data based on calculating the iterative moving average of a time series, in order to decompose the modelled and observed time series into contributions from different time

- scales. The Kolmogorov-Zurbenko filter is a low pass filter, with the length of the moving average window and the number of iterations determining the spectral component to be filtered. Taking the difference between two filtered time series (band-pass
  filter) makes it possible to decompose the observed and measured time series into an intra-diurnal component (ID, < 0.5 days),</li>
- a diurnal component (DU, 0.5-2.5 days), a synoptic component (SY, 2.5-21 days) and a long-term component (LT, >21 days) with the property

$$TS(x) = LT(x) + SY(x) + DU(x) + ID(x).$$
(3)

Here, TS describes the full time series of the species x. This is described in detail in Solazzo et al. (2017) and Solazzo and
Galmarini (2016) and references therein. Further detail is also given in the Supplementary Material.

By assessing the error of each component individually it is then easier to relate the error to the model process(es) characteristic at the respective time scale. The error analysis of the different spectral components is done by "error apportionment" (Solazzo et al., 2017), breaking down the mean square error (MSE) into bias, variance ( $\sigma$ ) error and minimum achievable mean square error (mMSE) as follows:

25
$$MSE = (mod - obs)^2 + (\sigma_{mod} - r\sigma_{obs})^2 + mMSE$$
(4)

As described by Solazzo and Galmarini (2016), the minimum achievable mean square error is determined by the observed variability that is not reproduced by the model. While this approach helps investigating the sources of model errors, it does not allow for clearly identifying or quantifying them as several processes take place on similar time scales, and because this filtering method does not allow for a complete separation of the different spectral components (see Solazzo et al., 2017, for a discussion of this issue).

30

In addition to this operational and diagnostic analysis of simulated NO2 concentrations, we include a brief evaluation of selected key meteorological parameters (temperature, wind speed and direction) as well as further chemical species (O3, NOx)<del>as , the former because</del> WRF-Chem is an online-coupled model<del>and , and the latter because</del> NO2 is tightly linked to NO and O3.

[revised manuscript text omitted]